# G2Sphere: Learning High-Frequency Spherical Signals from Geometric Data

## Abstract

Many modeling tasks from disparate domains can be framed the same way, computing spherical signals from geometric inputs, for example, computing the radar response or aerodynamics drag of different objects, or navigating through an environment. This paper introduces G2Sphere, a general method for mapping object geometries to spherical signals. G2Sphere operates entirely in Fourier space, encoding geometric structure into latent Fourier features using equivariant neural networks and then outputting the Fourier coefficients of the output signal. Combining these coefficients with spherical harmonics enables the simultaneous prediction of all values of the continuous spherical signal at any resolution. We perform experiments on various challenging domains including radar response modeling, aerodynamics drag prediction, and policy learning for manipulation and navigation. We find that G2Sphere significantly outperforms competitive baselines in terms of accuracy and inference time. We also demonstrate that equivariance and Fourier features lead to improved sample efficiency and generalization.

## 1 Introduction

Many different problems in 3D modeling require mapping detailed local geometric information to global spherical functions. In this paper, we consider several examples, such as simulating the radar response of an object, predicting the drag characteristics of a vehicle, or modeling an optimal control policy for manipulation. Modeling these types of signals is challenging because it requires very fine discretization of the domain in order to accurately capture high frequency information, such as sharp specular reflections in radar. This can make traditional numerical solvers and simulators expensive (Andersh et al., 2000). Deep learning methods represent an attractive, potentially more efficient approach.

One method of using neural networks to model spherical output signals is to output a tensor of regularly sampled values across the sphere using a spatial grid such as Healpix (Gorski et al., 2005) (Fig. 1 a). However, a large number of samples is necessary to attain a high spatial resolution over the sphere and even then, the model can easily overfit to the specific sample locations. Additionally,

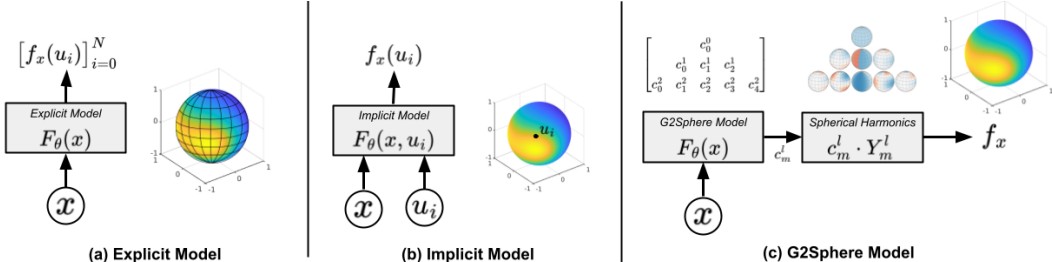

Figure 1: **Modelling Spherical Functions.** (a) The explicit model outputs a fixed grid of predictions to approximate the continuous function $f_x$. (b) The implicit model learns $f_x$ by conditioning on both input $x$ and coordinate point $u$. (c) G2Sphere decomposes $f_x$ into a set of learned Fourier coefficients $c_l^m$ and spherical harmonics $Y_l^m$. This enables prediction of $f_x$ at any coordinate point $u$.

these large input and output spaces can lead to other compounding issues such as training instability and computational costs (Vijayanarasimhan et al., 2014). An alternate strategy is to use implicit models which represent functions in a continuous manner and can be queried at any coordinate rather than outputting values at predefined coordinates (Fig. 1b). Implicit models have the advantages of being resolution independent, but they are relatively inefficient, requiring both a large amount of data and time to train in order to achieve good performance(Florence et al., 2022). Additionally, implicit models have been shown to be prone to overfitting to the training coordinates (Decugis et al., 2024) and, of particular interest to this work, they lose the geometric relationship between the input and output spaces.

In this work, we seek to address these challenges by introducing a new approach to mapping object geometries to spherical functions, where we model the output signal in Fourier space as a linear combination of spherical harmonic (SH) basis functions, enabling simultaneous prediction of the continuous output function at any desired resolution (Fig. 1c). While previous works have modeled mappings from geometric inputs to spherical functions using the spherical harmonics, they have relied on pre-defined mappings to convert the geometry to a spherical input (Esteves et al., 2018; Ha & Lyu, 2022). We propose G2Sphere, a novel approach to learn mappings between complex geometric inputs and spherical systems. G2Sphere maintains the geometric structure between input geometry and output signal by operating entirely in Fourier space and enforcing end-to-end SO(3)-equivariance. G2Sphere leverages equivariant graph convolutions (Geiger et al., 2022; Liao et al., 2023) to encode the geometry, spherical convolutions (Cohen et al., 2018) to render the spherical signal, and pointwise MLP non-linearities to model high-frequency information in the output space. This combination alleviates issues with existing equivariant GNN approaches which have been limited to lower frequency modeling, primarily due to computational constraints.

Our contributions are as follows:

- A novel SO(3)-equivariant architecture, G2Sphere, for mapping complex geometric inputs to high frequency, continuous spherical functions, utilizing Fourier decomposition. G2Sphere uses learned spherical embeddings for geometric inputs which are more general and flexible than prior works using engineered features.
- Empirical demonstration of the improved accuracy and efficiency of G2Sphere versus several baselines, across a wide range of challenging domains, including radar response and aerodynamic drag of 3D meshes, and policy learning for 2D and 3D navigation.
- Demonstration of zero shot super-resolution ability and generalization to novel meshes.
- In policy learning, an inference time of only 9ms compared to 156ms of diffusion policy, enabling our policy to achieve a high-frequency controller capable of running at 125Hz.

## 2 RELATED WORKS

**Fourier Transform in Machine Learning.** The Fourier transform (FT) is an important tool across a wide range of mathematical and engineering applications ranging from solving differential equations (Cooley et al., 1969) to quantum mechanics and signal processing (Arfken et al., 2011). Fourier transforms have also been heavily utilized in machine learning. In image processing and pattern recognition the FT has been used to speed up convolutional neural networks (Mathieu et al., 2013), to improve feature representation (Oyallon et al., 2018), and to increase the resolution of generative adversarial networks (Karras et al., 2021). Lee-Thorp et al. (2021) use the FT in the more modern transformer architecture to improve tokenization and improve performance. Li et al. (2020) combine neural operators with the Fourier transform to create the Fourier Neural Operator (FNO) to solve a number of challenging PDE systems and Bonev et al. (2023) extend this work to the Spherical FNO which they use to model weather systems. G2Sphere and Spherical FNO have a number of notable differences. First, G2Sphere operates entirely in Fourier space, whereas FNO moves back and forth between real and Fourier spaces. Additionally, FNOs are typically used to model dynamical systems which have similar continuous input and output spaces, while G2Sphere maps discrete geometric inputs to continuous outputs.

**Equivariant Neural Networks.** Equivariant neural networks constrain their layers to respect transformations under a symmetry group (Cohen & Welling, 2016; Geiger & Smidt, 2022; Weiler & Cesa, 2019). Neural networks equivariant to the 3D rotation group SO(3) are used for classifying

Figure 2: **G2Sphere Architecture.** Illustration of the G2Sphere (G2S) model. First, the geometric input is converted from real space (red dotted-line) to Fourier space (blue dotted line) using an SO(3)-equivariant encoder. We decode this spherical latent representation into a set of Fourier coefficients $c_m^l$ representing the continuous spherical signal. Finally, these coefficients are combined with a grid of pre-computed harmonic basis functions to map the function back to real space.

shapes (Esteves et al., 2018; Cohen et al., 2018), classifying protein structures (Weiler et al., 2018), and predicting features of atomic systems (Thomas et al., 2018; Brandstetter et al., 2021; Liao & Smidt, 2022). Similar to our method, all these approaches use steerable kernel bases and perform equivariant operations, such as convolutions or tensor products, but almost all are limited to harmonics of low degree ($\leq 10$). As such, G2Sphere is capable of capturing significantly more detail and fine-grained structure due to our use of a much higher maximum frequency ($\approx 40$). Esteves et al. (2018) and Cohen et al. (2018) model data over the sphere, but use a predefined mapping to convert the input into a spherical signal. They also focus on learning functions with low-dimensional discrete outputs, e.g. classification. Ha & Lyu (2022) also use a predefined mapping to convert brain geometry to spherical signals and a spherical UNet with frequency up sampling to do segmentation. Unlike these methods, G2Sphere uses a learned mapping to convert object geometries to spherical signals enabling the broad applicability of G2Sphere enabling the use the same model for a wide range of applications ranging from radar and aerodynamics to policy learning.

## 3 PROBLEM STATEMENT.

In this paper we consider the problem of mapping 3D geometric data, such as object meshes or points clouds, to *spherical signals* representing physical systems, such as radar response or drag. We consider spherical signals as continuous functions $f: S^d \to \mathbb{R}^c$ where $S^d$ is the 1-sphere or 2-sphere and $c$ is the number of channels in the signal. Given some input geometry $X \subset \mathbb{R}^3$, the objective is to learn a functional mapping $F: X \mapsto f_X$. Since full observations of $f_X$ are rare in practice, we assume a dataset consisting of only partially observed outputs. That is, a dataset $\mathcal{D} = \{(X_i, u_i, y_i)\}_{i=1}^N$ where samples consist of meshes or point clouds $X_i$, coordinates $u_i \in S^d$, and the value of the spherical signal $y_i = f_{X_i}(u_i) \in \mathbb{R}^c$ at $u_i$. Since the output signals are only partially observed, we consider generalization to new coordinates $u_i$ and new geometries $X_i$.

**Equivariance.** The value $f_X(u)$ may be considered as a property of the geometry $X$ defined with respect to viewing direction $u \in S^2$. We assume that both the geometry $X$ and direction $u$ are defined with respect to the same coordinate frame. Thus if the coordinate frame is rotated, both $X$ and $u$ are rotated, and $f_X(u) = f_{RX}(Ru)$ should be invariant. This property is equivalent to the equivariance of the functional mapping $F$; if $X$ is rotated to $RX$, the spherical signal $f_X$ is rotated to $f_{RX} = Rf_X$, preserving the relationship between the input geometry and the output signal. That is, $F(RX) = RF(X)$. This property can be enforced in the model architecture to ensure the learned model respects this property. Since the composition of equivariant functions is equivariant, it is sufficient to ensure that each component of the architecture is equivariant.

## 4 G2SPHERE

A G2Sphere model is comprised of two key components: (1) an encoder which compresses the input data into a compact latent representation and (2) a decoder which decomposes this latent representation into the spherical output utilizing spherical harmonics (Fig. 2). Encoder-decoder architectures such as U-Net (Zhou et al., 2018) and Transformers (Vaswani et al., 2017) are attractive methods for prediction tasks with high-dimensional inputs, such as 3D geometric data, due to latent space

compression. The encoder-decoder design is a good fit for our problem since we map between two different types of geometric data, e.g. mesh to spherical signals, and the encoder-decoder archetype allows us to match the geometry of the input and output spaces to the architecture of the encoder and decoder. By leveraging these learned and generalizable mappings from geometry to spherical signals, G2Sphere can be applied to tasks that were previously unattainable or computationally prohibitive under previous methods (Esteves et al., 2018; Cohen et al., 2018; Liao et al., 2023). We detail the concrete implementations utilized in our experiments in Appendix D.

## 4.1 ENCODER

In a G2Sphere model, the first step is to encode the local and global information of the 3D input into Fourier space, thereby ensuring Euclidean and permutation symmetries are correctly respected within our model. This is accomplished by utilizing an equivariant neural network encoder, but the exact architecture can vary depending on the type of input data. Broadly speaking, we define 3D geometric data as information that describes the shape, position, and spatial properties of objects in three-dimensional space. This can include various types of data including point clouds, voxel grids, object meshes, and RGB-D images. When dealing witth dense inputs, such as point clouds or object meshes, equivariant transformers such as Equiformer V2 (Liao et al., 2023) are used. However, when dealing with voxel grids or RGB-D images, $E(n)$-equivariant Steerable CNNs (Weiler & Cesa, 2019; Cesa et al., 2022) are a more natural choice. These types of models are constrained to use low-frequency spherical features, i.e. $l_{max} \leq 10$, due to the computational costs of higher-frequency features scaling significantly when applied on a per node basis. In each case, the input signal is initially spatially dispersed in $\mathbb{R}^3$ and at the end of the encoder, the signal is aggregated into a single feature vector representing the multi-channel spherical signal in Fourier space.

In Figure 2, we illustrate a G2Sphere model with mesh inputs which uses Equiformer V2 Liao et al. (2023), a $SO(3)$-equivariant transformer commonly used on 3D geometric data, as the encoder. The mesh is embedded as a geometric graph with node and edge features. The node feature for node $i$ is the position $\mathbf{x}_i \in \mathbb{R}^3$ and the edge features are the spherical harmonic embedding of the relative positions $e_{ij} = Y^l(\mathbf{x}_j - \mathbf{x}_i)$, where $Y^l \colon \mathbb{R}^3 \to \mathbb{R}^{2l+1}$ are the spherical harmonics and $i, j$ are vertices connected by an edge. Then a set of node-wise spherical features are learned over the graph which are mean pooled to form a multi-channel spherical latent space.

## 4.2 DECODER

Once the input has been projected into Fourier space using the encoder, we use operations that preserve the $SO(3)$ symmetry of the latent representation. Specifically, we use $SO(3)$-equivariant group convolutions to form a Spherical CNN (Cohen et al., 2018) decoder. The output of the final layer of the decoder $\hat{f}$ represents an $N$-channel signal over $S^2$ in the Fourier domain. That is, $\hat{f}_k = (c_{km}^l)_{m=0,l=0}^{2l+1,L}$ are the coefficients of the spherical harmonics up to frequency $L$ for $1 \leq k \leq K$. To convert the signal back to real space, we apply the inverse Fourier transform and evaluate these $K$ signals on the 2-sphere, at a position specified by spherical coordinates $(\theta, \phi)$. That is,

$$f(\theta, \phi) = \left( \sum_{l=0}^{L} \sum_{m=0}^{2l+1} Y_m^l(\theta, \phi) c_{km}^l \right)_{k=1}^{K},$$

where $Y_m^l$ are the spherical harmonics of degree $l$ and order $m$. In practice, we pre-compute the spherical harmonics basis functions at the desired grid resolution to allow for fast evaluation. See Appendix B for additional details on our harmonics implementations.

One drawback to using harmonics to approximate functions in this manner is that if the maximum frequency $L_{max}$ is low, then their resolution will also be low. The SH excel at capturing smooth, low-frequency components, but struggle with sharp features or discontinuous behavior unless a large number of harmonics are used, i.e. high $L$. As a result traditional equivariant decoder architectures, which operate with a relatively small number of harmonics ($L \leq 10$), struggle in domains where higher output fidelity is required. G2Sphere uses two techniques to improve output fidelity: *frequency up-sampling* where we gradually increase the frequency from the encoder's frequency $L_{enc}$ to the output frequency $L_{dec}$ and *trainable spherical non-linearities* (TSNL) (Bonev et al., 2023).

| Domain | Mesh Type | Transformer | Equiformer | Spherical CNN | G2S | G2S+TSNL |
|--------|-----------|-------------|------------|---------------|-----|----------|
| Radar | Frusta | $0.201 \pm$ 2e-4 | $0.271 \pm$ 3e-4 | $0.438 \pm$ 1e-6 | $0.221 \pm$ 1e-5 | **0.195 $\pm$ 9e-4** |
| | Asym | $0.179 \pm$ 5e-4 | $0.257 \pm$ 6e-4 | $0.496 \pm$ 1e-6 | **0.123 $\pm$ 2e-4** | $0.128 \pm$ 4e-4 |
| Drag | Pods | $0.064 \pm$ 1e-2 | $0.0672 \pm$ 4e-3 | | **0.062 $\pm$ 2e-3** | $0.064 \pm$ 6e-3 |

Table 1: **Mesh to Spherical Functions.** Mean square error of G2Sphere (G2S) and baselines on the radar and drag domains. Performance is averaged across 3 random seeds with $\pm$ standard error.

**Frequency Up-sampling.** Inspired by Ha & Lyu (2022), while increasing $L_{max}$ in the decoder, we gradually reduce the number of channels in our latent representation (See Appendix D). In order to maintain equivariance during this operation, we need to account for the relationship between the Fourier and real space. This is accomplished through the use of a regular non-linearity (as in Cohen et al. (2018) or De Haan et al. (2021)), where we map the signal to real space using the inverse Fourier transform (IFT), apply a point-wise non-linearity to the values of the spherical signal, and then perform the Fourier transform (FT), with higher frequency resolution, to convert back to the frequency domain. By combining a number of these non-linearities in our decoder, we are able to achieve a much higher maximum frequency (up to $L_{max} = 40$) than previous works using equivaraint architectures with dense geometric inputs, e.g. object meshes (Batzner et al., 2022; Kondor et al., 2018).

## 5 EXPERIMENTS

We systematically evaluate G2Sphere across three different domains: two supervised learning domains, where we learn a mapping from object meshes to spherical radar and drag signals, and a policy learning domain, where we learn a mapping between states and state-action values. These evaluations benchmark the performance of G2Sphere on dense (e.g., radar response) and sparse (e.g., drag coefficient) spherical signals and with complex (object mesh) and simple (object keypoint) input geometries. G2Sphere consistently outperforms the prior state-of-the-art in all domains. In the following, we provide an overview of each domain, our evaluation methodology, and our key findings.

### 5.1 RADAR AND DRAG MODELING FROM MESHES

Many modeling tasks take an object mesh as input and predict a physical property of that object. If the property depends on a direction vector, that is, a spherical coordinate $(\theta, \phi)$, then the output may be represented as a continuous function defined on the 2-sphere. We evaluate the performance of G2Sphere against baselines on two such tasks, predicting radar response and aerodynamic drag.

**Radar Prediction.** Due to the expense of collecting real-world radar data, we simulate radar responses for a variety of different meshes using a physical optics method to approximate the electromagnetic waves (Balanis, 2012). Specifically, we generate two mesh datasets, *Asym* and *Frusta*. See Fig. 11 for examples. The *Asym* dataset is a collection of randomly scaled basic shapes (cubes, cylinders, and spheres), with randomly generated protrusions to make the radar response asymmetric. The *Frusta* dataset represents a collection of complex 3D shapes, where each mesh is a combination of several basic components stacked together to form roll-symmetric objects. Due to this symmetry, we evaluate each frusta mesh over 360 orientations on the non-symmetric axis, $\theta \in [0, 2\pi]$ to generate the associated radar responses. The *Asym* meshes are evaluated over both axes, $\theta \in [0, 2\pi], \phi \in [0, \pi]$, where $\theta$ is discretized over 60 values and $\phi$ over 20.

**Drag Prediction.** We use a high-performance Computational Fluid Dynamics (CFD) simulator to generate a *Pods* dataset of mesh and drag coefficient samples of pod-like geometrical shapes. There are some differences relative to the radar dataset. First, as drag-coefficients are of most interest for flying objects (e.g. aircraft) the drag coefficients are only simulated for a cone in the front of the objects with $\theta, \phi \in [-20, 20]$. Secondly, a large contributing factor in the drag coefficients are the set of flight conditions (altitude, speed, etc.) which we treat as global parameters and append to the latent representation after encoding the mesh. The *Pods* dataset contains $10,000$ samples (mesh, flight conditions, drag) each with a single drag coefficient for a specific angle $(\theta, \phi)$, which defines

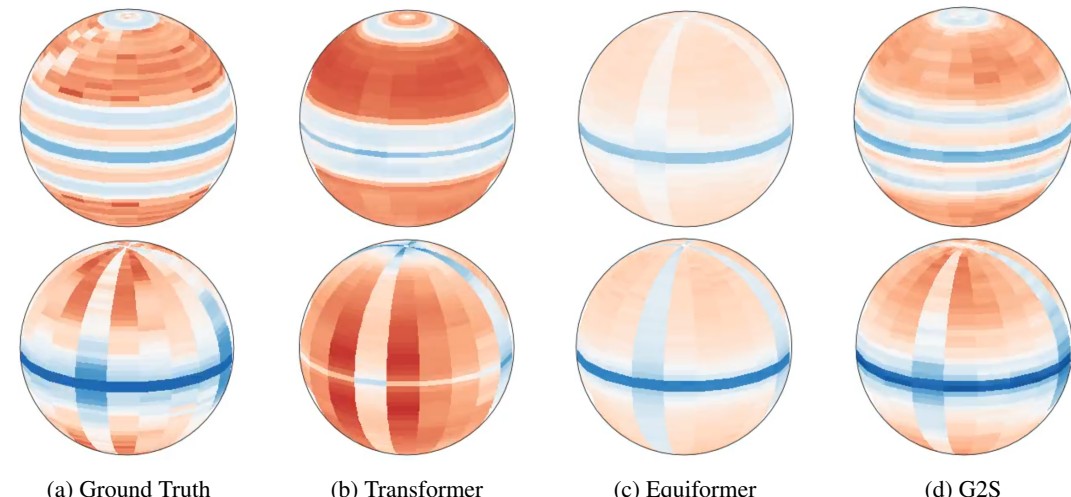

|  (a) Ground Truth | (b) Transformer | (c) Equiformer | (d) G2S |

Figure 3: **Radar Response.** Spherical outputs generated by the ground truth first-principles simulator, two baseline methods and G2Sphere. G2Sphere does a good job in reconstructing the overall projected shape characteristics and correctly capturing both types of radar scattering responses whereas the Transformer model underfits and the Equiformer model overfits.

the orientation of the incoming flow relative to the centerline of the Pod shape. Additional details on the radar and drag datasets can be found in Appendix F.

**G2Sphere Architecture.** We use Equiformer v2 (Liao et al., 2023) to encode the input mesh into a set of latent representations for each vertex in the mesh. These latent vectors are then combined into a single feature vector using a global average pooling layer. The decoder is a spherical CNN with several layers of spherical convolutions followed by the spherical non-linearity introduced in Sec. 4.2 to upsample the frequency from $L_{enc}$ to $L_{dec}$. We use $L_{enc} = 5$ for both tasks but use $L_{dec} = 40$ for the radar domain and $L_{dec} = 5$ for the drag domain. We examine two variants of G2Sphere, one with the trainable spherical non-linearity (G2S+TSNL), and one without (G2S).

**Baselines.** We compare the performance of G2Sphere against three competitive baselines for processing meshes, *Transformer*, *Equiformer*, *Spherical CNN*. The *Transformer* model is inspired by other prominent mesh-based transformer architectures (Siddiqui et al., 2024; Lin et al., 2021; Feng et al., 2018). It tokenizes the mesh into spatial and structural descriptors as in (Feng et al., 2018), and uses a transformer encoder with an MLP decoder to generate the predicted response. The *Equiformer* (Liao et al., 2023) model resembles the G2Sphere, but directly maps the latent representation to the discrete set of spherical values. The *Spherical CNN* (Esteves et al., 2018) model uses a ray-based method to map the input geometries onto the sphere, followed by a series of spherical convolutions. This can be considered a G2Sphere model where the learned encoder mapping is replaced with a predefined map. We use explicit models for the radar prediction task, where the baseline models output the full radar response for a given mesh. For the drag prediction task, we use implicit models where the coordinates are passed as input to the baseline models and appended onto the latent representation. This is necessary as, unlike in the radar domain, our drag dataset only has partially observed outputs and therefor lacks the dense targets required by explicit models. See Appendix D for additional information on model architectures and training details.

**G2Sphere Performance.** We report the performance of G2Sphere and the baselines on both radar and drag tasks in Table 1. G2Sphere obtains the lowest mean squared error compared to any of the baseline methods. Further, as the complexity of the task increases from the roll-symmetric *Frusta* shapes to *Asym* shapes, the difference in performance between G2Sphere and the baselines increases. This implies that G2Sphere is more adept at predicting dense output spaces than methods which output discrete values. While exact error rates for radar prediction vary depending on the task specifications, the Transformer baseline and G2Sphere fall within the most commonly cited 10 to 20% error range (Joseph T. Mayhan, 2024). Fig. 3 shows examples of the spherical radar signals generated by each model using meshes from the *Asym* dataset. When comparing the predictions generated by the equivariant models (Equiformer and G2Sphere), we can clearly see that G2Sphere

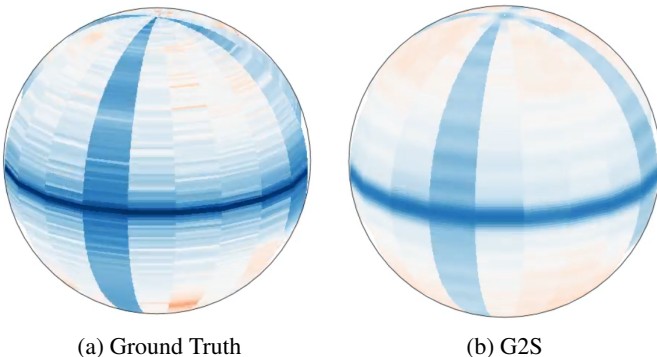

(a) Ground Truth           (b) G2S

Figure 4: **Zero-Shot Super-Resolution.** Radar response prediction on *Asymmetric Shapes* dataset with an G2Sphere model trained on $60 \times 20$ dataset and evaluated on $180 \times 20$ radar response samples.

captures significantly more detail and fine-grained structure which we attribute to the higher maximum frequency. Additionally, we can see that the Spherical CNN does very poorly, suggesting that the ray-based mapping from geometry to sphere does not capture the geometric information required for the radar prediction task. Similar to the radar task, we find the G2Sphere is able to achieve good performance on the drag task despite being trained on a sparse set of coordinate values. We note that all of our models achieve a error of around 6%, which falls within the single-digit percentage errors benchmark for aerodynamics (Naffer-Chevassier et al., 2024). In the following, we explore two interesting zero-shot capabilities of G2Sphere.

**Zero-Shot Super-Resolution.** G2Sphere outputs coefficients of Fourier basis functions and thus gives a continuous spherical signal which can be evaluated at arbitrary resolution. Therefore, G2Sphere can be trained on only low resolution radar data and evaluated at a higher resolution, i.e. a zero-shot super-resolution task. Fig. 4 shows an example where we train G2Sphere on $61 \times 21$ ($\theta \times \phi$) resolution radar data and transfer to $180 \times 21$ resolution. G2Sphere is the only model among the benchmarks (Transformer, Equiformer) that can do zero-shot super-resolution.

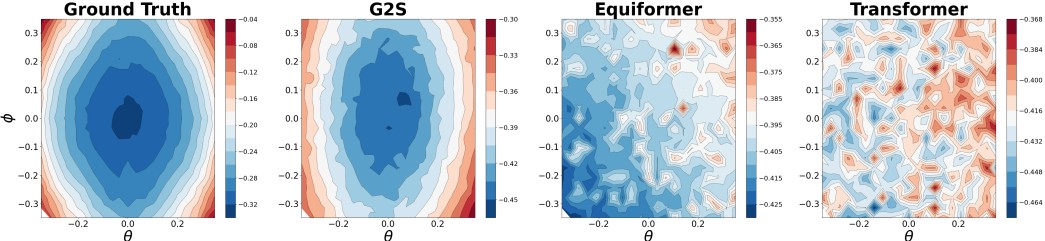

Figure 5: **Generalizing to Unseen Objects.** Full drag prediction on test sample not included in the *Pods* dataset. Unlike in our training dataset, we compute the entire drag prediction from $-20$ to $20$ degrees (shown in radians here). Despite being trained on single coordinate points, G2Sphere is able to reconstruct the general shape of the drag function, whereas the baseline implicit models overfit to specific points.

**Generalization to Unseen Object Geometries.** Our drag dataset contains only a single drag coefficient corresponding to a single $(\theta, \phi)$ for each object. In this experiment, we evaluate G2Sphere's ability to generalize to objects held-out during training and predict the full drag response cone, $(\theta, \phi) \in [-20, 20]$. Fig. 5 shows the generalization capability of G2Sphere when compared to the baselines. When compared to these implicit baseline models, G2Sphere demonstrates a stronger ability to generalize, effectively capturing the overall shape of the drag cone and producing more accurate predictions. Although all the baseline models perform well on the training data, here we see that they have overfit to specific $(\theta, \phi)$ coordinates, resulting in poor generalization to unseen regions beyond the training data.

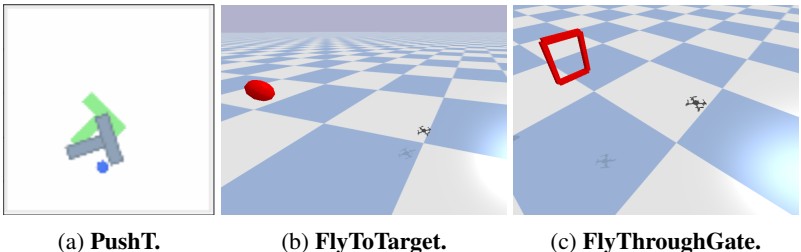

|                  |                     |                          |
|------------------|---------------------|--------------------------|
| (a) **PushT.**   | (b) **FlyToTarget.** | (c) **FlyThroughGate.** |

Figure 6: **Policy Learning Domains.** The tasks from our policy learning benchmarks. The fixed-goal variant of PushT is shown in (a). The target position to fly to is shown as a red ball in (b) and the gate to fly through is shown in red in (c).

## 5.2 POLICY LEARNING

In order to demonstrate the flexibility of G2Sphere for high-frequency continuous spherical signal learning, we apply G2Sphere on four policy learning tasks from two benchmarks (Florence et al., 2022; Panerati et al., 2021). Although there are many potential policy learning algorithms to which we could apply G2Sphere, in this work we explore the application of G2Sphere to behavioral cloning (BC) (Pomerleau, 1988). BC casts the policy learning task as a supervised learning problem where the agent learns to imitate a set of expert demonstrations. Following Florence et al. (2022), we formulate BC as a conditional energy-based modeling (EBM) LeCun et al. (2006) problem. EMBs define a policy through an energy function which assigns an energy value to each state-action pair. We compare G2Sphere to two prominent EBM policy learning methods, Implicit Behavior Cloning (IBC) (Florence et al., 2022) and Diffusion Policy (Chi et al., 2024), and demonstrate the G2Sphere outperforms these baselines both in terms of final performance and sample efficiency.

**Training Spherical Value Functions.** We use techniques from the energy-based model literature to train our G2Sphere EBM. Given a dataset of state-action samples $\{s_i, a_i\}$, where $s_i \in \mathbb{R}^N$ and $a_i \in \mathbb{R}^M$, we aim to learn an energy function $E : \mathbb{R}^{N+M} \to \mathbb{R}$. We train using contrastive learning (Le-Khac et al., 2020) by generating a set of non-expert actions $a'_j$ and utilizing an InfoNCE-style loss function (Oord et al., 2018) where the model must predict the true expert action $a_i$ from the non-expert actions $a'_j$.

The actions $a$ are 2D or 3D velocity vectors normalized to be in the unit ball. In order to model the energy function in Fourier space, we decompose $a$ into an action magnitude $\|a\|$ and an action direction $\hat{a} = a/\|a\| \in S^d$. For 2D actions, the energy function is evaluated

$$E_\theta(s, a) = \sum_{l,m} (F_\theta(s_i))^l_m B^l_m(a),$$

where $F_\theta$ is a G2Sphere model outputting Fourier coefficients $c^l_m$ and $B^l_m$ are polar harmonics (Appendix C), which give a basis of functions over the unit ball. For 3D actions, the magnitude is treated as an implicit variable and $E_\theta(s, a) = \sum_{l,m} (F_\theta(s_i, \|a\|))^l_m Y^l_m(\hat{a})$.

**G2Sphere Architecture.** Following Chi et al. (2024), we use keypoint observations which reduce the underlying geometries of objects in the scene to a set of keypoints that capture the spatial and structural information in the environment. Although we could still use an equivariant GNN encoder, given the small size and lack of variability of the input, an equivariant MLP is simpler. In the *PushT* domain, we have 2D keypoints and actions and therefore use $SO(2)$-equivariant linear layers. The *PyBullet Drones* domain is a 3D navigation task, and we use $SO(3)$-equivariant linear layers. The action magnitude is append to the latent representation after the keypoints are encoded. The decoder remains a spherical CNN but with a small modification for the 2D domains to perform convolutions on the 1-sphere. All our models use the same $L$ for both the encoder and decoder, that is, $L_{enc} = L_{dec} = 5$. To separate the impact of equivariance and representing the output in Fourier space, we consider a non-equivariant version of G2Sphere (NE-G2S) where we replace the equivariant MLPs with normal MLPs but still output the Fourier coefficients.

**PushT.** Adapted from Florence et al. (2022); Chi et al. (2024), the PushT task (Fig. 6a) requires pushing a T-shaped block (gray) to a fixed target (green) with a circular end-effector (blue). Object

| Domain | Task | IBC | E-IBC | Diffusion | NE-G2S | G2S |
|--------|------|-----|-------|-----------|--------|-----|
| PushT | Fixed | 0.85 (0.81) | 0.98 (0.95) | 0.95 (0.91) | 0.97 (0.95) | **1.0 (0.98)** |
| | Random | 0.62 (0.54) | 0.81 (0.76) | 0.71 (0.69) | 0.74 (0.70) | **0.92 (0.89)** |
| PyBullet Drones | FlyToTarget | 0.94 (0.87) | 0.98 (0.92) | 0.96 (0.94) | **1.00 (0.95)** | **1.00 (0.97)** |
| | FlyThroughGate | 0.90 (0.83) | 0.94 (0.87) | 0.94 (0.92) | 0.92 (0.90) | **0.98 (0.95)** |

Table 2: **Policy Performance.** Comparison of G2Sphere (G2S) against baselines on the PushT and PyBullet Drone domains. We compare max performance and, in parentheses, the average of last 10 checkpoints, each averaged across 50 different initialization conditions. G2Sphere significantly outperforms both an equivariant IBC model and the Diffusion Policy.

geometries are represented by a series of keypoints denoting the pose of the agent, block, and goal. We examine two variants: a fixed-goal variant where the target pose is always set to the pose in Fig. 6a and a randomized-goal variant where the target pose is randomly sampled within the workspace. The metric for performance is target coverage area between the block and the goal pose. We compare the performance of G2Sphere against two versions of IBC, standard (IBC) and equivariant (E-IBC), and Diffusion Policy. The results are shown in Table 2. We find that G2Sphere outperforms all other methods using the best checkpoints almost always achieves a perfect score in the fixed-variant. Additionally, we can see that the addition of equivariance stabilizes training instability leading to more consistent performance.

**PyBullet Drones.** We use the PyBullet Drones benchmark Panerati et al. (2021) to study the performance of G2Sphere in a 3D positional control domain. Specifically, we examine two tasks of varying difficulty, *FlyToTarget* (Fig. 6b) and *FlyThroughGate* (Fig. 6c). In the *FlyToTarget* task, the drone must take off from a landed configuration and fly to the target position. In the *FlyThrough-Gate* task, the drone is initialized in a flying configuration and must fly through a gate which is randomly posed around the drone. The observation space is composed of 4 keypoints for the drone, 4 keypoints for the gate, and a single keypoint for the target position. The action space is a delta motion from the current position which is passed to a PID controller. The agent operates at $30hz$ and the PID controller at $240hz$. The results are shown in Table 2. We see that similar to the PushT results, G2Sphere outperforms all other methods and provides increased training stability.

**Multimodality.** Learning from human demonstrations presents a significant challenge to behavior cloning due to the challenge of modeling the multimodal distributions common in human actions (Florence et al., 2022; Shafiullah et al., 2022; Hansen-Estruch et al., 2023; Chi et al., 2024). One advantage of G2Sphere is the ability to control the maximum multimodality of the policy via the maximum frequency. G2Sphere takes advantage of this by setting the maximum angular frequency to at least the amount of multimodality present in the task. We illustrate this behavior using the simple N-Paths task where the agent must navigate to the goal (green) using one of the N available paths. Fig. 7 shows that by setting the maximum frequency to the multimodality

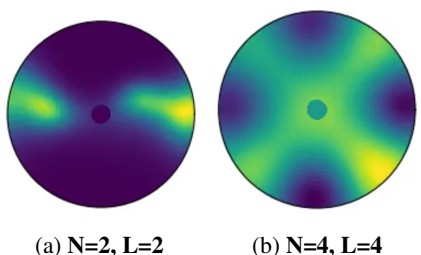

(a) **N=2, L=2**    (b) **N=4, L=4**

Figure 7: **Energy Landscape.** We visualize the energy landscapes generated with the G2Sphere on the N-Paths domain with $N = 2$ and $N = 4$, respectively.

present in 2-Paths and 4-Paths, the energy landscape generated by G2Sphere is able to model each of the multimodal trajectories. Additionally, Fig. 8 shows example trajectories generated by G2Sphere, E-IBC, and Diffusion. We note that G2Sphere is the best at matching the multimodality present in the task and is equally likely to take any of the N-paths, while diffusion shows bias towards certain paths and IBC fails to commit to any of the paths.

**Inference Speed.** It is critical to have a fast inference speed for many closed-loop real-time control tasks, such as robotic control. One of the known drawbacks to diffusion is that the denoising process requires a number of steps to optimize the trajectory. This is somewhat mitigated in the Denoising Diffusion Implicit Models (DDIM) approach(Chi et al., 2024), which has enabled relatively fast

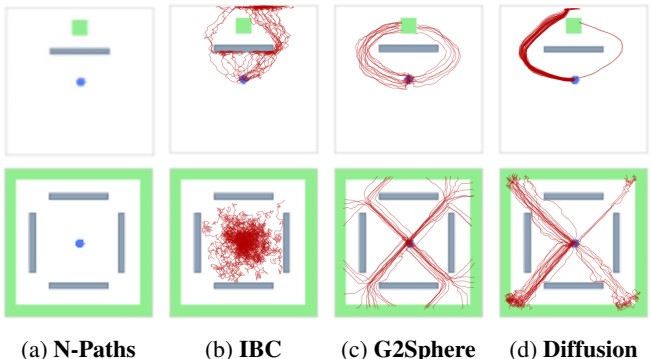

(a) **N-Paths**   (b) **IBC**   (c) **G2Sphere**   (d) **Diffusion**

Figure 8: **Multimodal Behavior.** From the start state (blue) there are two paths (top row) and four paths (bottom row) to the goal state (green). When using high enough angular frequency ($L = 2$, $L = 4$), G2Sphere learns all paths and commits to a path at the start of each rollout. Diffusion Policy exhibits a similar behavior but shows some bias towards certain paths whereas IBC fails to learn due to an equal distribution of expert path data. Actions are generated by rolling out 100 steps for the best preforming model.

| Domain | IBC | E-IBC | Diffusion | NE-G2S | G2S |
|---|---|---|---|---|---|
| PyBullet Drones | 32 | 83 | 156 | 3 | 9 |

Table 3: **Inference Speed (ms).** Comparison of inference speeds (ms) on an Nvidia Titan. IBC uses 3 iterations of derivative-free optimization and Diffusion utilizes 10 inference iterations to predict 8 actions. The speed reported for Diffusion is the inference time for 10 iterations divided by 8 to give the speed per action.

inference times of $100ms$. However, this does limit diffusion policies from many high-frequency control tasks, such as policies enabled with force-feedback. G2Sphere, on the other hand, has extremely fast inference time due to our use of grids of pre-computed harmonic function values, which can accurately predict the energy distribution in a single forward pass. Table 3 demonstrates the differences in inference speed for G2Sphere, IBC, and diffusion, with equivariant G2Sphere taking only $9ms$ compared to $156ms$ for Diffusion Policy.

## 6 LIMITATIONS AND DISCUSSION

In this work, we present G2Sphere, a novel approach to learning mappings from complex geometric inputs to continuous spherical signals. We evaluated G2Sphere across a wide range of challenging domains including radar response prediction, drag modeling, and policy learning for navigation and demonstrated improved performance on all tasks. Additionally, we demonstrated G2Sphere is capable of zero-shot super-resolution, full-resolution generalization to unseen geometries, the natural ability to model multimodality, and vastly improved inference times.

There are a number of limitations which future work could improve upon. First, in regards to the radar and drag domains, the output fidelity of G2Sphere is heavily dependent on the maximum harmonic frequency of the model. Unfortunately, current implementations of the spherical harmonics in equivariant network architectures result in an cubic increase in computational requirements as the harmonic frequency increases, thereby limiting the expressivity of our model. However, there do exist more efficient methods for calculating the spherical harmonics which would allow for increased maximum frequencies in future work (Wang et al., 2018; Schaeffer, 2013). Similarly, in this work we model the radial component of the spherical coordinates by either outputting a discrete set of spheres for different radii or by including it as an implicit variable. An alternate method would be to incorporate the radial component into the Fourier basis functions allowing for efficient computation of the all spheres continuously within some defined radial bounds (Wang et al., 2009). Finally, in this work we only explore 2D and 3D positional control policies and have omitted full 6DoF pose control. The model can be extended to 6DoF by applying inverse FT over SO(3) instead of $S^2$.

## 7 Reproducibility Statement

To ensure reproducibility, we have documented and made accessible all resources utilized by this work. Our source code, datasets, and visualization methods are provided for transparency. We have provided additional training and model architectural details for both G2Sphere and the various baselines in Appendix D. The techniques utilized to generate our high-quality synthetic data used in the experiments in Sec. 5.1 is detailed in Appendix F. Similarly, the details for the expert data used for policy learning experiments can be found in Appendix G alongside additional environmental details such as the sources for the environment implementations. Finally, the design decisions and implementation details of our harmonics implementation tailored to equivariant neural networks are disscussed in Appendix B.

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

## A  GROUP THEORY

**Equivariance.** A function is equivariant if it respects the symmetries of its input and output spaces. Specifically, a function $F\colon X \to Y$ is *equivariant* with respect to a symmetry group $G$, if it commutes with all transformations $g \in G$, $F(\rho_x(g)x) = \rho_y(g)F(x)$, where $\rho_x$ and $\rho_y$ are the *representations* of the group $G$ that define how the group element $g \in G$ acts on $x \in X$ and $y \in Y$ respectively. An equivariant function is a mathematical way of expressing that $F$ is symmetric with respect to $G$; if we evaluate $F$ for various transformed versions of the same input, we should obtain transformed versions of the same output.

**SO(3)-Equivariant Neural Networks.** For 3D physical systems, since the orientation of the coordinate frame is arbitrary, many task functions $f$ should be SO(3)-equivariant. For neural networks to incorporate this sort of geometric reasoning, it is necessary to parameterize signals $f\colon \mathcal{X} \to \mathbb{R}$, where $\mathcal{X} = S^2$ or $\mathcal{X} = \mathrm{SO}(3)$ in a way that is both computationally efficient and easy to apply rotations to. Cohen et al. (2018) provide an effective solution using the truncated basis of spherical harmonics $Y_m^l$ for signals defined over $S^2$ and Wigner D-matrix coefficients $D_{mn}^l$ for signals over SO(3). Writing $f\colon \mathrm{SO}(3) \to \mathbb{R}$ in terms of the $D_{mn}^l$ and then truncating to a given frequency $l \le L$ gives the approximation $f(g) \approx \sum_{l=0}^{L} \sum_{m=0}^{2l+1} \sum_{n=0}^{2l+1} c_{mn}^l D_{mn}^l(g)$. The SO(3) group convolution can be efficiently computed in the Fourier domain in terms of the coefficients $\{c_{mn}^l\}$ by convolution theorem. See Cohen et al. (2018) for additional details on the SO(3) group convolution.

**Circular and Spherical Harmonics** The *Circular Harmonics (CH)* describe functions on the 1-sphere (i.e. the circle $S^1$). They are solutions to Laplace's equation over $S^1$ defined $\psi_l(\theta) = e^{il\theta}$, where $\theta$ is the angular coordinate (Wang et al., 2009). In terms of group theory, the circular harmonics define irreducible representations for the 2D rotation group SO(2). Therefore, they are useful for constructing SO(2)-equivariant neural networks by representing circular signals as a combination of fixed CH basis functions and learnable CH coefficients (Weiler & Cesa, 2019).

The *Spherical Harmonics (SH)* extend the CH to describe functions on the surface of the 2-sphere $S^2$. Similar to the CH, they are orthonormal solutions of Laplace's equation on $S^2$, defined:

$$Y_m^l(\theta, \phi) = \sqrt{\frac{2l+1}{4\pi} \frac{(l-m)!}{(l+m)!}} P_m^l(\cos \theta) e^{im\phi},$$

where $\theta$ is the polar angle, $\phi$ is the azimuth angle, $l$ is the degree, $m$ is the order and $P_m^l$ is the associated Legendre function (Abramowitz & Stegun, 1948). The SH describe irreducible representations of the group SO(3) of 3D rotations. They are useful for creating SO(3)-equivariant features for SO(3)-equivariant deep learning (Geiger & Smidt, 2022).

## B    HARMONICS IMPLEMENTATIONS

In this section, we describe the details of our PyTorch harmonics packages which implements the Polar and Spherical Harmonics used in this work. While there are a number of existing spherical harmonics implementations (Bonev et al., 2023), they are tailored to the tasks that they solve, e.g. PDEs, and therefore do not integrate well with the equivariant neural networks used in this work (e.g. `e3nn` or `escnn`). Additionally, there are, to the best of our knowledge, no PyTorch implementations of the Polar Harmonics (detailed in Appendix C). To address these shortcomings, we developed our own differentiable implementations of the Polar and Spherical harmonics in PyTorch which we utilize throughout this work. The design goals of this package were two fold: (1) allow for the efficient integration of the harmonics with the output of `e3nn` or `escnn` and (2) enable the calculation of the harmonics on both a pre-defined grid of coordinates and at specific coordinates.

There are two considerations to make when tailoring our package to fit together with `e3nn` and `escnn`. First, these equivariant neural networks model the Fourier coefficients as the irreducible representations of the symmetry group and, therefor, our implementation needs to use the same representations. We note that this restriction makes a number of previous implementations such as Bonev et al. (2023) a poor fit as they use different derivations of the harmonics with different Fourier coefficient specifications. Secondly, because these equivariant models typically operate in real space, we need the harmonics to be in real space as opposed to complex. For example, we use the sin and cos form of the circular harmonics (Appendix C) to match the dimensionality of the $SO(2)$ irreducible representations.

The second goal of our package is enable efficient batch computation of the harmonics over many values. We require the ability to both compute the harmonics at specific coordinates (commonly used during training to match the partial views $u$ in our dataset) *and* to pre-compute a grid of harmonics values at pre-defined coordinates (typically used at inference time to decrease computation time). In order to achieve this, when instantiating the harmonics, a grid of harmonics values are pre-computed at the desired resolution and a number of intermediate variables are saved during this process. These intermediate variables are specific to the type of harmonics but, in general, they are components of the harmonics independent to the coordinates such as the normalization constants and Fourier mode components. This pre-computation allows for the efficient evaluation of the harmonics and reduces training times.

## C    POLAR FOURIER TRANSFORM

In this section we derive the Polar Harmonics Basis functions used in our 2D policy learning experiments by performing Fourier analysis on the Polar coordinates. We would ideally like for these functions to be decomposed into radial and angular components such that we can view this decomposition as an extension of the normal Fourier transform. We can fulfill this requirement by requiring the basis functions to take the separation-of-variables form:

$$f(\rho, \phi) = P(\rho)\Phi(\phi). \tag{1}$$

### C.1    BASIS FUNCTIONS

The angular component of the basis function is simply:

$$\Phi_l(\phi) = \frac{1}{\sqrt{2\pi}} e^{im\phi}, \tag{2}$$

where $l$ is an integer. However, this definition resides in the complex domain, and when using equivariant neural networks, it is advantageous to be in the domain of real numbers. Therefore, we will instead use the sine-cosine form:

$$\Phi_l(\phi) = \frac{1}{\sqrt{2\pi}} \big( \cos(l\phi) + \sin(l\phi) \big), \tag{3}$$

where $l \geq 0$. The associated transform in angular coordinates is simply the normal 1D Fourier Transform.

We will rely on Bessel functions to match the functions of arbitrary spatial frequency required by the radial component of the Polar basis function. Using the Bessel function as a basis gives rise to the $l$-th order Fourier-Bessel series:

$$P_{lm}(\rho) = \frac{1}{\sqrt{N_{lm}}} J_l(k_{lm}\rho), \tag{4}$$

where $N_{lm}$ is a normalization constant and $k_{lm}$ is the $k^{\text{th}}$ zero of $J_1$.

## C.2 POLAR FOURIER TRANSFORM

By combining these radial and angular components we can form the complete basis function for the polar Fourier transform:

$$\Psi_{lm}(\rho, \phi) = P_{lm}(\rho)\Psi_l(\psi), \tag{5}$$

where $P_{lm}$ is defined by Eq. 4 and $\Psi_l$ by Eq. 2. As $\Psi_{lm}$ forms an orthonormal basis on the region $\rho < a$, we refer to it as the *Polar Harmonics*. A function defined over polar coordinates, $f(\rho, \phi)$, can be expanded using the polar Fourier transform with the Polar basis function, $\Psi_{nm}$, and Polar coefficients, $P_{lm}$:

$$f(\rho, \phi) = \int_0^{\inf} \int_0^{\inf} P_{lm}\Psi_{lm}(\rho, \psi)m dm, \tag{6}$$

where

$$P_{lm} = \int_0^{\inf} \int_0^{2\pi} f(\rho, \phi)\Psi_{lm}^*(\rho, \phi)\rho d\rho d\phi. \tag{7}$$

While this infinite transform is of theoretical interest, in practice we instead use the transform defined on a finite region, e.g. the unit-circle. That is, a function $f(\rho, \phi)$, where $\rho \leq a$ can be defined as:

$$f(\rho, \phi) = \sum_{m=1}^{\inf} \int_{l=0}^{\inf} P_{lm}\Psi_{lm}(\rho, \psi), \tag{8}$$

where

$$P_{lm} = \int_0^a \int_0^{2\pi} f(\rho, \phi)\Psi_{lm}^*(\rho, \phi)\rho d\rho d\phi. \tag{9}$$

# D TRAINING DETAILS

In this section we provide additional details about the model architectures and training process for the experiments in Sec. 5. All models were trained until convergence measured by loss in the supervised learning domains and policy performance in the policy learning domains. All of our experiments are run on Nvidia Xeon-g6-volta GPUs on a high-performance cluster.

## D.1 MESH-TO-SPHERE

**Transformer.** The Transformer architecture is composed of a mesh face embedding layer, transformer encoder, and an MLP decoder. The mesh face embedding layer takes individual mesh faces as an input and uses geometric features like vertex position and normal direction to calculate an embedded representation of the face. The application of this layer to the mesh generates a sequence of embedded faces; a learned classification embedding is prepended to this sequence. A standard transformer encoder as introduced in Vaswani et al. (2017) is then applied to the embedded sequence to transfer global information about the mesh faces to the classification embedding; after this operation, the MLP decoder is used to convert the classification embedding into the radar response. We train using the Adam optimizer (Kingma, 2014) with the best learning rate and its decay were chosen to be $5e^{-4}$ and $1e^{-5}$ respectively. We use a batch size of 4.

**G2Sphere.** For the mesh-to-sphere domain, the G2Sphere model uses Equiformer v2 to encode the object mesh into Fourier space. We use the equiformer architecture from Liao et al. (2023) with 4 layers where each layer a feature dimension of 128, a $l_{max}$ of 5, and 4 attention heads. We then use a global mean pooling head to compress the node features returned by Equiformer into a single latent representation of size 128. The decoder is a 5 layer Spherical CNN Cohen et al. (2018)

with the feature dimensions $[128, 64, 32, 16, K]$, where $K$ is the number of output channels in the spherical signal, and $l_{max}$ dimensions $[5, 10, 15, 20, 40]$. We use the ReLU activation in our SO(3)-activation layers in the Spherical CNN. We train using the Adam optimizer (Kingma, 2014) with the best learning rate and its decay were chosen to be $1^{-4}$ and $0.95$ respectively. We use a batch size of 4.

**Equiformer.** The Equiformer baseline has the same encoder architecture as G2Sphere but a slightly different decoder architecture. The Spherical CNN decoder in Equiformer, has 6 layers where the feature dimensions are $[128, 64, 32, 16, K]$, where $K$ is the number of output channels in the spherical signal, and $l_{max}$ dimensions $[5, 5, 5, 5, 5]$. We use the ReLU activation in our SO(3)-activation layers in the Spherical CNN. At the end of the Spherical CNN, a final spherical convolution converts the latent representation into the explicit grid of output values. We use the ReLU activation in our SO(3)-activation layers in the Spherical CNN. We train using the Adam optimizer (Kingma, 2014) with the best learning rate and its decay were chosen to be $1^{-4}$ and $0.95$ respectively. We use a batch size of 4.

### D.2 Policy Learning

**G2Sphere.** In the 2D and 3D policy learning domains, our observations are a set of 2D/3D keypoints which describe the objects in the scene. Therefore, we use much simpler models than in the mesh-to-sphere domain. Specifically, we use SO(2) and SO(3)-equivariant MLPs as both our encdoers and decoders. In the *PushT* domain, we use a 4-layer $SO(2)$-equivaraint MLP for our encoder with a 512-dimensional latent representation and a 4-layer, 512-feature dimension, $SO(2)$-equivariant spherical CNN with as the decoder. This $SO(2)$ spherical CNN is essentially a series of spherical convolutions on the 1-sphere as opposed to the 2-sphere. Both encoder and decoder use a $L_{max}$ of 3. Similarly in the *PyBullet Drones* domain, we use a 4-layer $SO(3)$-equivaraint MLP for our encoder with a 256-dimensional latent representation and a 4-layer, 256-feature dimension, spherical CNN decoder. Both encoder and decoder use a $L_{max}$ of 5. Both models use a dropout Srivastava et al. (2014) of 0.1 while training. We train using the Adam optimizer (Kingma, 2014) with the best learning rate and its decay were chosen to be $1^{-4}$ and $0.95$ respectively. We use a batch size of 256 and train our contrastive loss using 256 negative samples.

**IBC.** We use two IBC models in our policy learning experiments: a non-equivariant version which uses standard MLPs and a equivariant version which uses SO(2) or SO(3)-equivariant MLPs. The non-equivaraint IBC is comprised of a 4 layer MLP encoder and a 4 layer MLP decoder both with 1024 feature dimensions. The implicit actions are appended onto the latent representation after it has been encoded. The equivariant version of IBC, has the same structure as the non-equivariant model but uses a feature dimension of 512 and a $L_{max}$ of 3 and 5 (for the 2D and 3D domains respectively). We train using the Adam optimizer (Kingma, 2014) with the best learning rate and its decay were chosen to be $1^{-4}$ and $0.95$ respectively. We use a batch size of 256. The contrastive loss is generated using 256 negative samples following Florence et al. (2022). We use a dropout Srivastava et al. (2014) of 0.1 while training. At inference time, we perform 3 iterations of Derivative-Free Optimization (Florence et al., 2022) using 4096 samples to select the best action.

**Diffusion.** We use the implementations from Chi et al. (2024) for our Diffusion models, specifically, we use the transformer-based architectures. The transformer has 8 layers with 4 heads where each token is embedded as a 256 feature vector. We use the Square Cosine Schedule proposed in iDDPM (Nichol & Dhariwal, 2021) which is cited as the best performing noise scheduler in Chi et al. (2024). We train using the Adam optimizer (Kingma, 2014) with the best learning rate and its decay were chosen to be $1^{-4}$ and $0.95$ respectively. We use a batch size of 256. We use 100 denoising diffusion iterations for both training and inference. At inference time the model predicts the next 16 actions of which 8 are executed.

## E Maximum Frequency Ablations

A unique capability of G2Sphere is its ability to leverage the maximum harmonic frequency L to control bias of the model. A higher L captures more fine details and higher-frequency components of the underlying function which can lead to a more complex and accurate approximation. Therefore when trying to predict the high-frequency radar response function, we find that the higher the L

| **Asym** | $L_{dec} = 5$ | $L_{dec} = 10$ | $L_{dec} = 25$ | $L_{dec} = 40$ |
|---|---|---|---|---|
| G2S | 0.383 | 0.218 | 0.148 | **0.123** |

Table 4: **$L_{dec}$ Ablation.** Mean square error of G2Sphere using increasing maximum output harmonic frequencies. As the output frequency of the decoder increases the performance of the G2Sphere model improves. Performance is averaged across 3 training seeds.

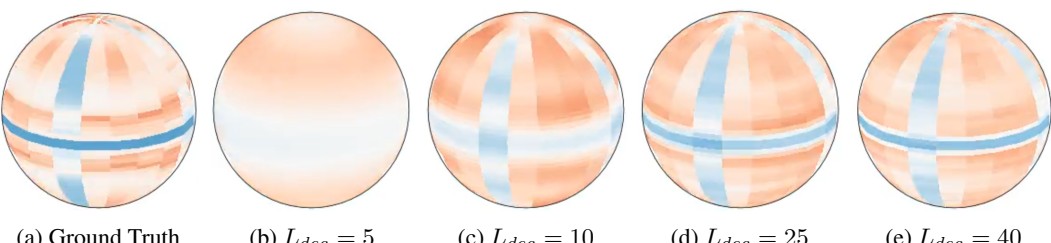

(a) Ground Truth    (b) $L_{dec} = 5$    (c) $L_{dec} = 10$    (d) $L_{dec} = 25$    (e) $L_{dec} = 40$

Figure 9: **Effect of $L_{dec}$ on Fidelity.** As the maximum frequency of the decoder increases so does the accuracy of the G2Sphere prediction. At $L_{dec} = 5$ we see that most of the features of the signal are lost and we are only capable of modelling the general high/low signals. However, as we increase to $L_{dec} > 5$ we can see the features getting sharper as the output frequency increases.

the more accurate our predictions become (Table 4, Fig. 9). However, we note that this is only possible due to the large scale radar datasets which have responses over the entire sphere for each mesh. In contrast, in domains where we lack these dense outputs, such as the drag and policy learning domains, we find a lower L can combat overfitting by providing a smoother, lower-detailed approximation of the underlying function (Fig. 10).

# F   DATASET GENERATION

## F.1   FIRST-PRINCIPLE RF MODEL

Due to the scarcity of available real-world and simulated radar data for training radar models, we simulate our own benchmark dataset. To generate ground truth data, we use the physical optics approximation method (Balanis, 2012), which provides a linear approximation of the more general and highly non-linear scattering formulation for electromagnetic waves. A simple operator that describes physical optics response across the illuminated section of an object for perfectly reflecting material as,

$$F(x, k) = \frac{ik}{2\pi} \int_{R^3} e^{-i2k\langle x, y \rangle} dy,$$

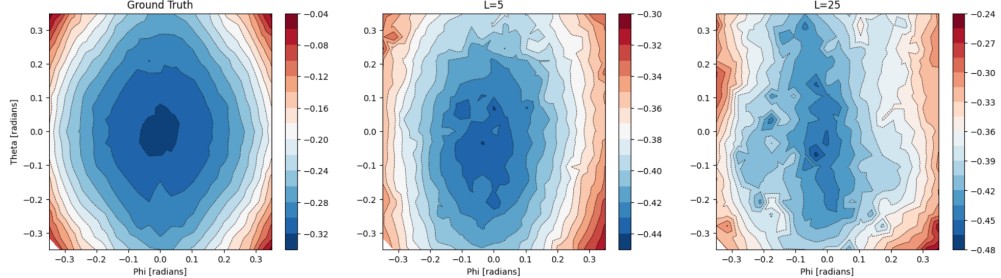

Figure 10: **Effect of $L_{dec}$ on Generalization.** As the maximum frequency of the decoder increases, G2Sphere starts to overfit to the sparse data samples in the Drag domain. As a result, the G2Sphere model with a lower output frequency generalizes better to new data.

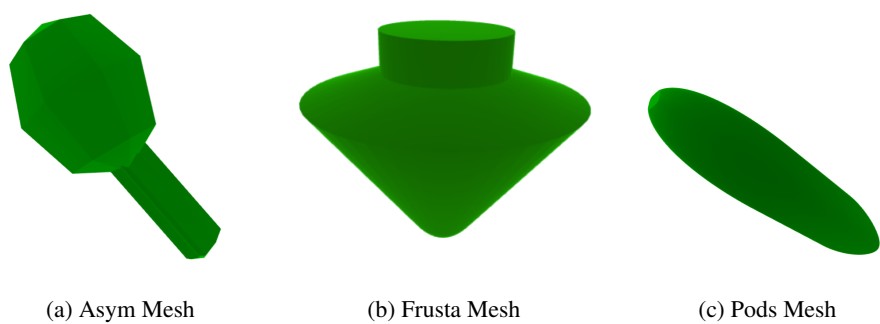

(a) Asym Mesh        (b) Frusta Mesh        (c) Pods Mesh

Figure 11: **Object Meshes.** Example object meshes used in the mesh-to-sphere experiments. The Asym and Frusta meshes are used in the radar prediction domain and the pods meshes are used in the aerodynamics prediction domain.

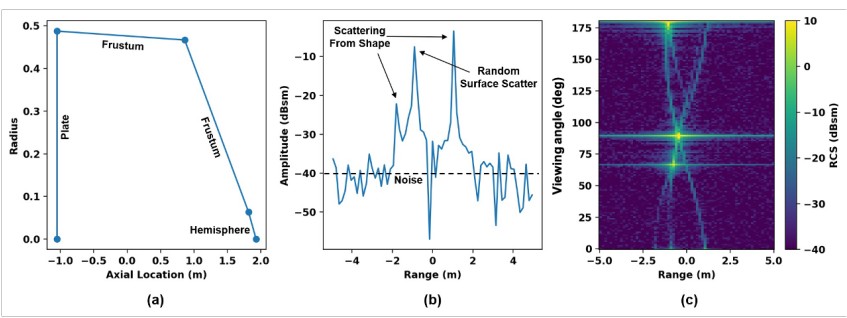

Figure 12: **Radar Simulator.** (a) Cross section of random 3D object mesh. (b) The range profile for $20°$ viewing angle. (c) The corresponding static radar pattern.

where the incident wave number is $k = 2\pi/\lambda$, $\lambda$ is the wavelength, and the observation unit vector is,

$$x = (\sin\theta\cos\phi, \sin\theta\sin\phi, \cos\theta),$$

for $\theta \in [0, \pi]$ and $\phi \in [0, 2\pi]$. We note that the approximation is valid only in high frequency regions such that $k << 2\pi/D$, where $D$ is the length of the longest side of the object.

As we are particularly interested in far-field sensing, the physical optics approximation is useful as a fairly accurate and flexible simulation tool for training data generation. The simulation input is a parameterized mesh object (an example cross section shown in Fig. 12(a)). The simulation calculates the the physical optics response for given observation line-of-sight and frequency and the total radar response of the object is equal to the sum of the individual triangle responses that are visible to the radar. Simulations for this work required generating the response across a linear set of frequencies to emulate a Linear Frequency Modulated (LFM) waveform, where the center frequency is $3e9$Hz and bandwidth is $4e8$Hz using circular polarized waves with orientation RL.

The Radar Cross Sec. (RCS) for each triangle is calculated using legacy software (Burt & Moore, 1991). The simulation produces a radar observation, $r \in \mathbb{R}^{N_r}$ (Fig. 12(b)) for a given viewing angle. The observation is the normalized magnitude of the range-profile as described in Sec. III of Chance et al. (2022). All the normalized range profiles are then stacked across varying radar viewing angles to generate what is referenced as a radar static pattern (Fig. 12(c)). Note that fixing the radar line-of-sight and rotating the object would generate the same radar static response.

We utilize two different mesh datasets for our experiments. The first is the *Frusta* dataset. Each object in this dataset is defined by a series of stacked frusta objects, in which adjacent frusta share the same radius to ensure a continuous object. To generate a diverse set of objects, we vary both the number of frusta components and the radial parameters for each frusta. To introduce additional object and radar response diversity, we also vary the ends of the frusta object to either be flat planes, half spheres, or one of each.

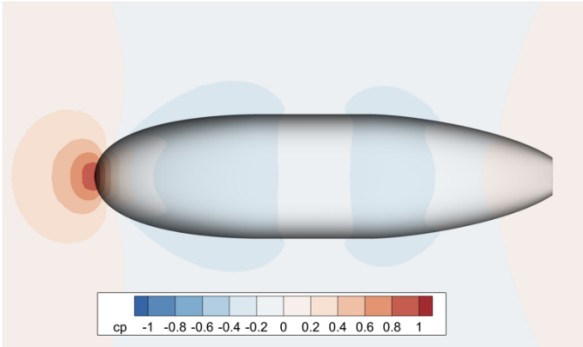

Figure 13: **Aerodynamics Simulator.** Ground truth drag coefficients generated by the numerical CDF simulator.

While the *Frusta* dataset provides objects with complex forms, it is also rotational symmetric along the roll axis. To address this issue, we also generate the *Asym*. This dataset consists of three different base shapes; cubes, cylinders, and spheres. Each object is randomly scaled across it's independent dimensions; to ensure diversity, no random scaling is repeated within object class. After the scaling, the mesh is duplicated and subdivided to decrease the relative size of mesh faces. A group of adjacent faces is selected at random from the overall set of mesh faces on the subdivided copy; these faces are then extruded by a random amount in the average normal direction of the selection. This extruded protrusion is then added to the original mesh using the union operation, resulting in an object without any rotational symmetries.

Using these methods, we generate a dataset of X objects for the *Frusta* dataset, and 30,000 objects for the *Asym* dataset, evenly distributed across the underlying classes. Each dataset is divided 90/10 between train/validation and test sets. Since the meshes resulting from the Frusta generation method are large relative to the capacity of neural network models (maximum size of 4500 faces), frusta meshes are decimated to 50% of their face count using quadratic decimation for training and testing.

### F.2 Aerodynamic Simulation and Data

The aerodynamic dataset consists of "pod-like" shapes defined parametrically by varying shape parameters such as: length, diameter, bluntness, and cross-section asymmetry using Latin Hypercube Sampling. Flight conditions including altitude, Mach number and Reynolds number were also selected using Latin Hypercube Sampling, where altitude ranges from 0 to 50 kilofeet, Mach number ranges from 0.05 to 0.5, and Reynolds number ranges from $10^6$ to $5 \times 10^8$. For each shape permutation a high-quality computational grid is algorithmically generated. The minimum cell size is defined to provide over 100 cells across the length of each shape, and 50-100 volume cells to resolve the boundary layer down to the viscous sublayer, i.e. $y^+ = 1$, where $y^+1$ is the wall coordinate commonly used in defining the law of the wall. Generation of drag force data is based on a Computational Fluid Dynamics (CFD) simulator for which we solve the Reynolds-Averaged Navier-Stokes (RANS) mass, momentum and energy equations through the NASA FUN3D simulation framework. Turbulence is modeled using the Spalart-Allmaras model with a freestream turbulence intensity of 3 percent. The van Albada flux limiter with the low-diffusion flux splitting "LDFSS" flux construction method. Simulations are run until a relative residual tolerance of $10^{-5}$ is met for the mass, momentum, energy, and turbulence governing equations.

The Pods dataset contains $10,000$ samples of (mesh, flight conditions, drag) tuples, each with a single drag coefficient for a specific angle $\theta, \phi \in [-20, 20]$, which defines the orientation of the incoming flow relative to the centerline of the Pod shape.

## G  POLICY LEARNING ENVIRONMENTS

**PushT.** We generate 100 expert demonstrations for both the fixed and randomized PushT tasks. All demonstrations were generated by a single human operator familiar with the task. Each episode terminates either after the task is completed or the maximum number of steps (500) is reached. We use the PushT implementation found in Chi et al. (2024) which can be found here: `https://github.com/real-stanford/diffusion_policy`.

**PyBullet Drones.** We generate 50 expert demonstrations for both the *FlyToTarget* and *FlyThrough-Gate* tasks. All demonstrations were generated by an expert policy which has access to the underlying simulation state. We add a small amount of uniformly sampled noise, $\epsilon \sim U(0,1)$, to each action from the expert policy. Each episode terminates either after the task is completed or the maximum number of steps (300) is reached. We use the PyBullet Drones implementation found in Panerati et al. (2021) which can be found here: `https://utiasdsl.github.io/gym-pybullet-drones/`.

## H  EXAMPLE RADAR RESPONSE OUTPUTS

In Fig. 14 we highlight representative examples generated by the ground truth simulator, transformer, equiformer, and G2Sphere on the Frusta dataset. We note that although G2Sphere still produces the best predictions, the difference between G2Sphere and the transformer baseline is much less pronounced on this dataset. This aligns with the result in Table 1 which shows that the performance of the transformer baseline and G2Sphere to be similar.

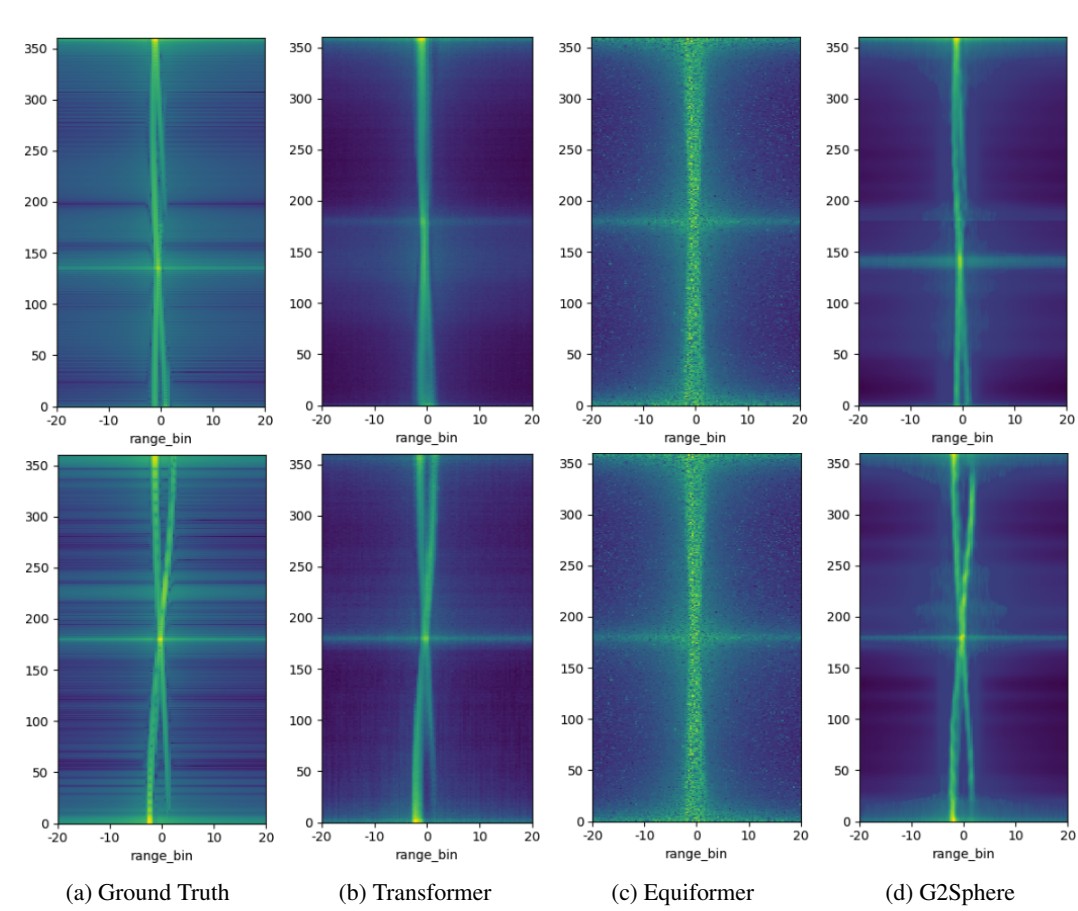

(a) Ground Truth      (b) Transformer      (c) Equiformer      (d) G2Sphere

Figure 14: **Frusta Radar Predictions.** Radar response predictions on the Frusta dataset. Due to the roll symmetry in the Frusta dataset we plot the spherical outputs as static patterns. G2Sphere is best able to capture both the overall structure of the response and also the speculars (bright horizontal lines).

