# OpenReview forum: "G2Sphere: Learning High-Frequnecy Spherical Signals From Geometric Data"
_ICLR.cc/2025/Conference — Submitted to ICLR 2025_

### Official Review · Reviewer_1JJ3 · 2024-11-02

**Soundness:** 3
**Presentation:** 2
**Contribution:** 2
**Rating:** 5
**Confidence:** 3

**Summary:**

This paper presents G2Sphere, an approach to model spherical signals by mapping geometric 3D data, such as meshes or point clouds, to spherical representations using Fourier coefficients. The model addresses tasks involving spherical data, which are essential for applications like radar response and aerodynamic drag modeling. The authors propose an architecture composed of an SO(3)-equivariant encoder (using Equiformer v2) to encode 3D geometric data and a spherical CNN decoder to predict Fourier coefficients, allowing the reconstruction of high-resolution spherical signals.

**Strengths:**

- The paper introduces a Fourier-based approach to represent spherical signals, making it relevant for domains where rotational consistency and high-frequency detail are crucial.
- By adequately modified the exisiting model’s architecture which respects rotational equivariance, the approach maintains input-output alignment under rotations, which is beneficial for tasks that require orientation-aware outputs.

**Weaknesses:**

- The paper’s explanation of the model architecture is somewhat ambiguous, particularly regarding the encoder-decoder structure. Specifically, the description of the Equiformer encoder, its output domain, and the role of mean pooling is unclear. For instance, it is not explicitly described how the mesh is encoded into the latent space or how this representation transforms in the decoder to produce Fourier coefficients. From my understanding, G2Sphere applies node-wise spherical representation in Equiformer’s intermediate layer and then combines these node-wise features into a single spherical signal through pooling. However, this flow is difficult to follow based on the paper's explanation alone.
- The proposed method lacks novelty, as modeling the coefficients of spherical signals directly for continuous function representation has been explored in other domains [1]. Specifically, this paper's approach seems to mainly add equivariant GNNs and equivariant convolutions for modeling spherical coefficients, which limits the originality of its contribution. Additionally, a similar frequency up-sampling technique has been used in another spherical harmonics-based convolution method [2].
- While the model is designed to be equivariant, the paper would benefit from a straightforward proof or brief explanation of how this equivariance is ensured within the network architecture. Providing this clarification could help readers better grasp the model's theoretical foundation
- The paper could enhance clarity by including more details on the training process, particularly the training algorithm and loss functions. This would enable readers to understand the optimization approach better and assess the model's robustness.
- Highlighting G2Sphere's uniqueness and distinct purpose would strengthen the paper’s impact. Similar tasks involving spherical signal modeling exist, such as Implicit Neural Networks on spheres [3,4,5] or spherical convolutions for brain imaging [2]. Differentiating G2Sphere from these related approaches would underscore its novelty and relevance.

[1] https://arxiv.org/abs/2311.10908

[2] https://ieeexplore.ieee.org/stamp/stamp.jsp?tp=&arnumber=9759394

[3] https://openreview.net/forum?id=g6UqpVislvH

[4] https://openreview.net/forum?id=Y5SEe3dfniJ

[5] https://arxiv.org/abs/2402.05965

**Questions:**

- Could the authors clarify the output domain of the Equiformer encoder and the overall training process? This clarification would greatly enhance the reader’s understanding of the model.

---

> ### Author Response · Authors · 2024-11-22
>
> Thank you for your review. We are working on a revision to incorporate your feedback, but we address some of your concerns and questions here as well.
>
> ---
>
> **Model Architecture Clarity:**
>
> *“The paper’s explanation of the model architecture is somewhat ambiguous, particularly regarding the encoder-decoder structure. Specifically, the description of the Equiformer encoder, its output domain, and the role of mean pooling is unclear. [...]. From my understanding, G2Sphere applies node-wise spherical representation in Equiformer’s intermediate layer and then combines these node-wise features into a single spherical signal through pooling. However, this flow is difficult to follow based on the paper's explanation alone.”*
>
> Your understanding of the encoder is largely correct. The encoder does learn node-wise spherical features which are then pooled to form our spherical latent space. This may be considered as a signal spherical function, but note it is multi-channel $S^2 \to \mathbb{R}^k$. The decoder then uses a series of spherical convolutions to reduce the channels of the spherical features while increasing the maximum frequency of the spherical representations. We will revise the model section to make this more clear.
>
> ---
>
> **Novelty:**
>
> *“The proposed method lacks novelty, as modeling the coefficients of spherical signals directly for continuous function representation has been explored in other domains [1]. Specifically, this paper's approach seems to mainly add equivariant GNNs and equivariant convolutional for modeling spherical coefficients, which limits the originality of its contribution. Additionally, a similar frequency up-sampling technique has been used in another spherical harmonics-based convolution method [2].”*
>
> Thank you for the references, although we were aware of some of these works we did not spend enough time discussing them in relation to our work. We have added additional details and cited a number of these suggested works.  We believe that the pairing of an equivariant GNNs encoder and spherical CNN decoder is a significant and novel contribution.  Despite a large number of works applying equivariant GNNs to various geometric signals, all are limited to relatively low maximum frequency ($ \leq 11$).  The spherical CNN decoder allows our model to produce much higher frequency output signals.
>  The breadth of applications considered in our experiments ranging from supervised learning to policy learning, demonstrates the value of an architecture designed to map from geometry to spherical signals. We note that these applications are novel and that we achieve improved performance on alldomains.
>
> ---
>
> **Equivariance Proof:**
>
> *“While the model is designed to be equivariant, the paper would benefit from a straightforward proof or brief explanation of how this equivariance is ensured within the network architecture. Providing this clarification could help readers better grasp the model's theoretical foundation”*
>
> This is a good idea. We have added additional details demonstrating the equivariance relationship between the input object geometries and the output spherical functions.
>
> ---
>
> **Training Details:**
>
> *“The paper could enhance clarity by including more details on the training process, particularly the training algorithm and loss functions. This would enable readers to understand the optimization approach better and assess the model's robustness.”*
>
> These details are in Appendix D including the training process of both G2S and the baseline methods. We will make this clearer in the revision.
>
> ---
>
> **G2Sphere's uniqueness and distinct purpose:**
>
> *“Highlighting G2Sphere's uniqueness and distinct purpose would strengthen the paper’s impact. Similar tasks involving spherical signal modeling exist, such as Implicit Neural Networks on spheres [3,4,5] or spherical convolutions for brain imaging [2]. Differentiating G2Sphere from these related approaches would underscore its novelty and relevance.”*
>
> Thank you for the additional works around modeling spherical functions, we have added these to our related works and have included additional comments differentiating our method from these. Broadly speaking these methods all deal with data that is already spherical in nature. [3,4,5] all examine domains with spherical inputs and [2] uses a predefined mapping to convert the brain into a spherical representation. Our method is an end-to-end trained method which learns this mapping between the input geometries and the spherical functions and is therefore more broadly applicable to new domains. Additionally, [2] is performing a segmentation task and therefore the input and output spaces are very similar. In contrast, our method is concerned with mapping between two very different types of data, e.g. geometric input and spherical signal outputs.

---

> ### Comment · Reviewer_1JJ3 · 2024-11-22
>
> Thank you for your thoughtful clarification. I appreciate the additional insights you have provided. However, after further reflection and considering the overall contributions and impact of the paper, I will stand by my original score.
>
> Furthermore, I must honestly admit that I found it challenging to fully assess the value of this task and the impact of your experimental results. I refrained from raising the score primarily because the work does not seem to align with the standards of novelty or efficiency typically expected in the areas of Equivariance and Geometrical Modeling. However, I would strongly recommend emphasizing why this task is important and how your work addresses limitations or enables capabilities that were previously unattainable. Doing so could significantly strengthen the case for your contributions.

---

> ### Author Response · Authors · 2024-11-25
>
> Emphasizing the tasks is a great suggestion. We will edit the paper to clarify the following points.  Namely, we believe that outside the usual spherical and equivariant GNN benchmarks there is a rich set of real world tasks mapping 3D geometry to high-frequency spherical signals. Beyond the radar, drag, and policy tasks in the paper, grasp affordance prediction, object pose recognition, part assembly, and physics-based simulation are a few more examples.
>
> Our paper shows that current models underperform on these tasks (output limited in frequency, poor generalization, etc.)  One reason is that existing equivariant spherical approaches have primarily relied on predefined task-specific mappings to transform geometric data into spherical representations[1,2,3]. While effective in narrow contexts, these mappings lack generality and are unsuitable for tasks requiring adaptability across varying geometries and objectives. For instance, value function approximation with basis functions has been explored in prior work [7], but its adoption has been limited due to inefficiencies and reliance on handcrafted features. We highlight this application for two reasons: (1) policy learning is a domain where efficiency is a critical component and we demonstrate in Table 2 that G2S outperforms very strong baseline methods including other equivariant architectures and (2) our work represents a significant step forward in this field of Fourier value function approximation.
>
> Alternatively, existing equivariant GNN approaches have been limited to lower frequency modeling, primarily due to computational constraints [4,5,6]. Depending on the application, effective and practical solutions would be required to be efficient (e.g., policy learning) and provide high fidelity reconstructions (e.g., radar and aerodynamic modeling). Similarly, for the radar simulation application, the input shape complexity and variation (e.g., shape asymmetry), the orientation dependency, and the high frequency characteristic of the output signal ($L_{max} > 50$) make current off-the shelf, explicit techniques struggle with achieving the level of fidelity reconstruction required to replace computationally expensive first-principle methods.
>
> Our method is directly motivated by the limitations of current methods on these tasks.  By leveraging generalizable, learnable mappings from geometry to spherical signals, our work facilitates tasks that were previously unattainable or computationally prohibitive under previous methods.
>
> ---
>
> [1] Esteves et al. Learning SO(3) Equivariant Representations
> with Spherical CNNs, https://arxiv.org/pdf/1711.06721
>
> [2] Ha et al. SPHARM-Net: Spherical Harmonics-Based Convolution for Cortical Parcellation, https://ieeexplore.ieee.org/document/9759394
>
> [3] Cohen et al. Spherical Convolutions, https://arxiv.org/pdf/1801.10130
>
> [4] Liao et al. EquiformerV2: Improved Equivariant Transformer for Scaling to Higher-Degree Representations, https://arxiv.org/abs/2306.12059
>
> [5] Thomas et al. Tensor field networks: Rotation- and translation-equivariant neural networks for 3D point clouds, https://arxiv.org/abs/1802.08219
>
> [6] Brandstetter et al. GEOMETRIC AND PHYSICAL QUANTITIES IMPROVE
> E(3) EQUIVARIANT MESSAGE PASSING, https://arxiv.org/pdf/2110.02905
>
> [7] Konidaris et al. Value Function Approximation using the Fourier Basis, https://people.csail.mit.edu/gdk/pubs/fourier-msrl.pdf

---

### Official Review · Reviewer_9Htm · 2024-11-04

**Soundness:** 2
**Presentation:** 2
**Contribution:** 3
**Rating:** 6
**Confidence:** 3

**Summary:**

The authors present an architecture for mapping geometric datatypes to spherical output maps. To do so they use existing techniques from spherical architectures in combination with an equivariant encoder which maps mesh data to spherical signals. The paper is written clearly and for the most part it is easy to understand. A diverse set of example is provided to understand the efficacy and relevance of the method.

The paper has a few weaknesses in terms of the experimental evaluation of the method and in references made to other works which utilize similar methods already. Moreover some of the claims made in the paper are not supported by experimental data and/or references.

**Strengths:**

Interesting architecture which uses a mapping from 3d geometries to spherical domain. Diverse set of examples are shown, with seemingly good results over existing baselines

**Weaknesses:**

The main two weaknesses I see are:
- 125-127: the authors claim that the architectuere caprtures significantly more detail than existing architectures but this is never experimentally shown.
- at some points references to prior works are missing - especially with respect to certain techniques in the architecture which have been utilized in the architecture before (See details)
- The experiments are not-well motivated and it is not clear to me 1) how relevant these are and 2) how difficult and fair they are wrt existing baselines. It is hard to quantify how good the errors reported in Table 1 are. For instance, [1] uses a similar approach to map geometry to spherical signal. Why not compoare wrt. this baseline which seems better suited.
- It is hard to understand the method and the experimental performance with the main text alone. I suggest adding a better explanation of the architecture.

**Detailed:**
- the signals to be learned in FIgure 3 show a high dependence on l and very little variation in m. As such I worry that this example is biased towards a specific architecture. Why not reconstruct other spherical Signals?
- 230-239: **TSNL** -trainable spherical non-linearities are not a new concept as this has also been used in [2]
- 215-229: again, applying non-linearities in spatial domain and then going back to frequency domain has been previously done. See e.g. [1,2,3]
- error is only reported in terms of MSE. As far as I understand the error is not properly integrated over the sphere using the jacobian?
- 100: FNO/SFNO do not entirely operate in Fourier space.
- Table 1 reports numerical errors but it is hard to evaluate how good these errors are. In particular given that Figure 5 suggests that the Equiformer and Transformer break down completely.

[1] https://arxiv.org/pdf/1711.06721
[2] https://arxiv.org/abs/2306.03838
[3] https://arxiv.org/abs/2312.05225

Edit: Raising score from 5 to 6.

**Questions:**

225-229: While this seems like a substantial improvement, L=40 is not particularly high. For instance in the SFNO work that you cite, SH coefficients are computed up to degrees ~360. Why is this architecture not able to do so?
742: you claim that previos implementations use different derivations of the Spherical Harmonics which makes them a poor fit. Can you elaborate?

---

> ### Author Response · Authors · 2024-11-22
>
> Thank you for your review. We are working on a revision to incorporate your feedback, but we address some of your concerns and questions here as well.
>
> ---
>
> **Novelty:**
>
> *“At some points references to prior works are missing - especially with respect to certain techniques in the architecture which have been utilized in the architecture before”*
>
> Thank you for the references. We have added them to the paper and clarified our contribution in relation to them. We view the main novelty of our architecture to be the fact that it combines an equivariant GNN mesh encoder and a spherical CNN decoder, improving on previous methods utilizing equivariant GNNs which are highly constrained in the frequency of their output signal. Moreover, we emphasize the broad applicability of G2S which enables us to use the same model for a wide range of applications ranging from radar and drag to policy learning. We note that these applications are novel and that we achieve improved performance on all domains.
>
> ---
>
> **L vs M:**
>
> *“The signals to be learned in FIgure 3 show a high dependence on l and very little variation in m. As such I worry that this example is biased towards a specific architecture. Why not reconstruct other spherical Signals?”*
>
> If we understand your comment correctly, you are concerned that our model architecture is biased towards the radar responses in our Asym-Shapes dataset. We would like to note a couple things here. First, the lack of variance in m, is primarily due to the meshes used  to generate the radar responses. These meshes are roll-symmetric objects with added protrusions to make the responses asymmetric resulting in the horizontal banding you see in Fig. 3. The vertical bands (Fig. 3 bottom) are due to radar scatters from the asymmetric protrusions. Secondly, G2S uses all orders (l) and degrees (m) based on the $l_max$ used. Therefore, our architecture is not biased to any type of spherical signal. Because of this, G2S represents a more general method which we can easily apply to other domains. In fact our other domains (drag and policy learning) are examples of these types of signals and our model performs well there as well. This is different from other approaches in the spherical modeling domain like SFNO which uses $l_max$ and $m_max$ parameters. Please reply if we misunderstood your comment and question. We are happy to discuss more.
>
> ---
>
> **Model Architecture Clarity:**
>
> *“It is hard to understand the method and the experimental performance with the main text alone. I suggest adding a better explanation of the architecture.”*
>
> We are working on revising the model section to make the architecture more clear. In particular, we are adding additional details to highlight how the input geometries are encoded into the spherical space. Would you mind pointing out some of the areas which you found confusing so we can make sure these are addressed?

---

> > ### Comment · Reviewer_9Htm · 2024-11-23
> >
> > I thank the authors for their detailed response.
> >
> > Regarding the radar dataset - would it make sense to include shapes which do not have this symmetry? It seems like a more interesting test case. Moreover, I would like to also point out here that SFNO is a mapping from functions on the sphere to functions on the sphere, as opposed to the geometry-to-spherical signal setting that you have here. For the prior,  spherical convolutios make a lot of sense, which automatically removes the dependency on $m$ for the filter functions.
> >
> > Regarding model clarity - I welcome the effort on improving the manuscript on this front. For me, the main uncertainty is in how input geometries are processed. The current figure does not reveal that.

---

> > > ### Author Response · Authors · 2024-12-03
> > >
> > > **“Regarding the radar dataset - would it make sense to include shapes which do not have this symmetry? It seems like a more interesting test case.”**
> > >
> > > Yes it would and we are very interested in exploring this in the future. Due to the radar domain, the objects in our current datasets exhibit symmetries present in flying objects, e.g. airplanes, due to aerodynamics. However, we are currently looking at expanding this to more complex shapes. In particular, we are interested in taking some models from ShapeNet and generating their radar responses.
> > >
> > > ---
> > >
> > > **“Regarding model clarity - I welcome the effort on improving the manuscript on this front. For me, the main uncertainty is in how input geometries are processed. The current figure does not reveal that.”**
> > >
> > > Thank you for clarifying, we will make sure these details are included in the revision. At a high level, the input geometry is converted into a graph with SO(3) features, i.e. points in SO(3) and edge lengths. You could think of this as a SO(3) embedding of sorts. Then you can use whatever SO(3) equivariant encoder architecture, we use Equifromer v2 but any equivariant graph network should work.

---

> ### Author Response · Authors · 2024-11-22
>
> **Experiments & Baselines:**
>
> *“The experiments are not-well motivated and it is not clear to me 1) how relevant these are and 2) how difficult and fair they are wrt existing baselines. It is hard to quantify how good the errors reported in Table 1 are. For instance, [1] uses a similar approach to map geometry to spherical signal. Why not compare wrt. this baseline which seems better suited.”*
>
> The experiments demonstrate the wide range of applications for which can be formulated as  learning a map from object geometries to spherical functions and solved using G2S. Each experimental domain showcases an important feature of the G2S method. The Mesh-to-Radar experiments demonstrate that G2S can learn high-resolution spherical functions and can outperform other explicit baselines. The Mesh-to-Drag experiments compare G2S to implicit models and demonstrate the improved generalization capability of G2S. Finally, the policy learning experiments demonstrate the sample efficiency of G2S and provide us with a number of very strong baselines (i.e. IBC and Diffusion) to compare against.  Thank you for suggesting [1].We are currently running experiments to add [1] to our set of baselines. These results will be in the revision and we will post them here as well once they have finished training. This will let us assess the importance of using a learned mapping between object geometries and spherical signals (as opposed to the predefined mapping in [1]).
>
> In terms of the mesh-to-sphere baselines, we would highlight the transformer model as a strong baseline to compare against. Transformer-based models, e.g. MeshGPT [5] and MeshGraphormer[6], have shown impressive performance encoding geometry inputs. Additionally, our transformer uses a more complex tokenization of the mesh including additional features such as the surface normals. On the Frusta dataset, for example, the transformer model performs very well and comes close to G2S. Similarly, Equiformer has been used for similar mesh-based learning tasks but not typically for high resolution predictions.
>
> The performance of our model can also be evaluated relative to the acceptable errors for applications in these domains. In Table 1, we point out that the MSE reported for G2S and Transformer fall within the generally acceptable range of error for radar prediction [4]. Equiformer does perform poorly which we attribute to the low $l_max$ required by this method. Similarly, in drag prediction a 5-10% error is normally considered acceptable [3] which our models fall within (e.g. 6.2% for G2S and 6.5% for the Transformer). We would like to point out that the generalization experiment in Fig..5 is a very challenging task, and typically implicit models do poorly on it. These models are only trained on a single data point for each mesh so generalizing to the full space is quite hard.
>
> ---
>
> **Higher Res Predictions:**
>
> *“125-127: the authors claim that the architecture captures significantly more detail than existing architectures but this is never experimentally shown.”*
>
> Our claim that G2S captures significantly more detail than other existing architectures (125-127) is backed by our experimental results. For example, the comparisons to Transformer and Equiformer demonstrating this relationship, see Figure 3. Note that the G2S model captures the most detail in the horizontal banding in the top example and the most detail along the vertical band in the bottom example. This claim is tied into the fact that G2S has a higher maximum frequency than other equivariant mesh-to-sphere methods which allows it to capture more detail in high-resolution predictions. As we previously mentioned we are adding a baseline inspired by [1] which will allow us to see if the higher output frequency or if the learned mapping from mesh to sphere are the more important component of this improved prediction.

---

> > ### Comment · Reviewer_9Htm · 2024-11-23
> >
> > higher-res predictions: Perhaps I am missing something here, but wouldn't it be more fair to add additional maximum frequencies to the other approaches? It is true that equivariant approaches typically suffer due to existing implementations not being particularly efficient and therefore do not reveal how well they work due to computational limitations. However this is precisely where stronger inductive biases would be expected to have a benefit.
> >
> > The aforementioned fix in e3nn should be easy as there exist implementations which use a more efficient algorithm, see i.e. https://github.com/NVIDIA/torch-harmonics/blob/main/torch_harmonics/legendre.py

---

> > > ### Comment · Reviewer_9Htm · 2024-11-23
> > >
> > > I thank the authors for taking the time to answer my questions and addressing my concerns. I would still suggest the authors  think about ways to improve the experimental section to better convince readers of their method. For the remaining issues I am positive that the authors will address my concerns in the manuscrupt, and I have decided to raise my score to 6.

---

> > > ### Author Response · Authors · 2024-12-03
> > >
> > > **“Perhaps I am missing something here, but wouldn't it be more fair to add additional maximum frequencies to the other approaches? ...**
> > >
> > > We agree that this would be a more fair comparison and are interested in exploring this in the future. However, we would push back a bit on how easy it would be to implement this. The feature representations, i.e. the learned SH coefficients, in e3nn/escnn are created such that the symmetry groups can be combined, restricted, etc. See https://quva-lab.github.io/escnn/api/escnn.group.html#representations for more details. In practice, this means that it's not as simple as just changing the manner in which the SH are calculated.

---

> ### Author Response · Authors · 2024-11-22
>
> **Maximum Frequency:**
>
> *“​​225-229: While this seems like a substantial improvement, L=40 is not particularly high. For instance in the SFNO work that you cite, SH coefficients are computed up to degrees ~360. Why is this architecture not able to do so?”*
>
> While high maximum frequencies are commonly used in works dealing with spherical signals, high frequency outputs are not common for equivariant GNNs conditioning on  object geometries as input. The geometric input has two effects: (1) increased computational requirements and (2) reliance on the implementation of  SHs in equivariant GNN frameworks such as eSCN and e3nn. These frameworks use the simplest derivation of the SH (which have an exponential increase in compute as $l$ increases) unlike SFNO which reduces the computational requirements by computing the projection onto the associated Legendre polynomials via quadrature and the projection onto the harmonic functions via the FFT [2]. Additionally, our encoder learns spherical features for each point in the input, i.e. the mesh, which is computationally expensive even when using a low $L_{max}$. Due to these two restrictions, we can only achieve a maximum output frequency of $L_{max}=40$ on a v100 GPU. In future work, we may consider improving the efficiency of SH harmonic implementations in e3nn to attain even higher frequencies.
>
> *“742: you claim that previous implementations use different derivations of the Spherical Harmonics which makes them a poor fit. Can you elaborate?”*
>
> The comment about other derivations of the SH being a poor fit is primarily due to the feature representations used by these equivariant NNs (see previous answer for more details). Because of this, the output SH coefficients from our equivariant architecture are in a particular format which we cannot use with other implementations of the SH, e.g. torch-harmonics from SFNO.
>
> ----
>
> **Minor Notes:**
>
> *“100: FNO/SFNO do not entirely operate in Fourier space.”*
>
> You are correct. We were attempting to point out that FNO/SNFO are not applicable to our geometry-to-sphere domain. We have modified our explanation.
>
> *“Error is only reported in terms of MSE. As far as I understand the error is not properly integrated over the sphere using the jacobian?”*
>
> Thank you for pointing this out. We will fix this and report our results correctly.
>
> ---
>
> [1] Esteves et al. Learning SO(3) Equivariant Representations
> with Spherical CNNs, https://arxiv.org/pdf/1711.06721
>
> [2] Schaeffer, Efficient Spherical Harmonic Transforms aimed at
> pseudo-spectral numerical simulations, https://arxiv.org/pdf/1202.6522v2
>
> [3] Naffer-Chevassier et al. Enhanced Drag Force Estimation in Automotive Design: A Surrogate Model Leveraging Limited Full-Order Model Drag Data and Comprehensive Physical Field Integration, https://www.mdpi.com/2079-3197/12/10/207
>
> [4] Mayhan et al. Measurement-based radar signature modeling, https://mitpress.mit.edu/9780262048118/measurements-based-radar-signature-modeling/
>
> [5] Siddiqui et al. MeshGPT: Generating Triangle Meshes with Decoder-Only Transformers, https://arxiv.org/abs/2311.15475
>
> [6] Lin et al. Mesh Graphormer, https://arxiv.org/abs/2104.00272

---

### Official Review · Reviewer_i2M4 · 2024-11-05

**Soundness:** 2
**Presentation:** 3
**Contribution:** 2
**Rating:** 5
**Confidence:** 4

**Summary:**

The authors introduce a novel neural network architecture to learn spherical signals from 3D geometric data. Their network maps geometric inputs to fourier coefficients in an equivariant graph convolutional network. The learned coefficients are used in a spherical neural network decoder amended with a novel frequency up-sampling technique to produce a continuous spherical signal of arbitrary precision. The proposed approach outperforms baselines on radar prediction, drag prediction and policy learning.

**Strengths:**

The problem considered is interesting and relevant to the community. The paper is well written and organized, and the figures are useful for understanding the proposed approach and its contributions.

**Weaknesses:**

The architectural contribution appears fairly minimal, much of the architecture is a combination of existing methods; however, the proposed frequency up-sampling method and learned nonlinearity appear novel. Of the novel architectural contributions only the frequency up-sampling method seems to provide consistent empirical benefit.

The application domain appears fairly novel. The authors introduce new datasets for radar prediction and drag prediction which extends their contribution. Their results on these datasets exceed the predictive performance of baseline models considerably; however, it is unclear to me if the selected baselines are the most appropriate for these tasks.

**Questions:**

Is the model appropriate for weather radar or tornado prediction[1]? There are several existing datasets in these domains and numerous domain specific baselines that could be compared against.

[1] https://arxiv.org/pdf/2401.16437

---

> ### Author Response · Authors · 2024-11-22
>
> Thank you for your review. We are working on a revision to incorporate your feedback, but we address some of your concerns and questions here as well.
>
> ---
>
> **Novelty:**
>
> *“The architectural contribution appears fairly minimal, much of the architecture is a combination of existing methods; however, the proposed frequency up-sampling method and learned nonlinearity appear novel. Of the novel architectural contributions only the frequency up-sampling method seems to provide consistent empirical benefit.”*
>
> The novelty of our model is exactly that it is the first to combine an equivariant GNN mesh encoder and a spherical CNN decoder. A relatively large number of papers apply equivariant GNNs to tasks defined on geometric graphs or meshes, but almost all are limited to harmonics of low degree ($\leq 11$).  The spherical CNN circumvents this limitation.  Moreover, we emphasize the broad applicability of G2S which enables us to use the same model for a wide range of applications ranging from radar and aerodynamics to policy learning. We note that these applications are novel and that we achieve improved performance on all domains. Within the policy learning domain, G2S represents a significant innovation in method. This concept of approximating value functions using basis functions was first introduced in [4], but its applicability was limited to simple tasks, e.g. cart-pole, mountain climber etc. Our method allows for the broad application of this style of value function modeling via basis functions. The value of TSNL is indeed task dependent and hinges on whether the SH up to $L_{max}$ are sufficient to model the output signal.
>
> ---
>
> **Experiments & Baselines:**
>
> *“The application domain appears fairly novel. The authors introduce new datasets for radar prediction and drag prediction which extends their contribution. Their results on these datasets exceed the predictive performance of baseline models considerably; however, it is unclear to me if the selected baselines are the most appropriate for these tasks.”*
>
> As noted, since the domains are novel, we worked to establish strong and appropriate baselines from related domains in the literature. For the radar and drag tasks, the transformer model is a strong and reasonable baseline.
> The Transformer model is inspired by other prominent mesh-based transformer architectures [2][3][4]. It tokenizes the mesh into spatial and structural descriptors as in [4] and uses a transformer encoder with an MLP decoder to generate the predicted response. Additional details on the transformer baseline can be found in Appendix D.1. Similarly, Equiformer has been used for similar mesh-based learning tasks but not typically for high resolution predictions. That being said, we are currently running experiments to add an additional method to our baselines which is inspired by the original spherical CNN work [1]. In this method they use a predefined mapping to convert meshes spherical signals as opposed to our method which uses a learned mapping via the GNN encoder. These results will be in the revision and we will post them here as well once they have finished training. This will let us assess the importance of using a learned mapping between object geometries and spherical signals (as opposed to the predefined mapping in [1]).
>
> ---
>
> **Minor Notes:**
>
> *“Is the model appropriate for weather radar or tornado prediction[1]? There are several existing datasets in these domains and numerous domain specific baselines that could be compared against.”*
>
> This is an interesting idea. However, our method is designed around converting object geometries to spherical signals, and we do not believe that it is applicable to the weather or tornado domains you cite. This type of problem is applicable to Fourier modeling (e.g. FNO & SFNO), but our emphasis is on the relationship b/t the geometry and the spherical signals.
>
> ---
>
> [1] Esteves et al. Learning SO(3) Equivariant Representations
> with Spherical CNNs, https://arxiv.org/pdf/1711.06721
>
> [2] Siddiqui et al. MeshGPT: Generating Triangle Meshes with Decoder-Only Transformers, https://arxiv.org/abs/2311.15475
>
> [3] Lin et al. Mesh Graphormer, https://arxiv.org/abs/2104.00272
>
> [4] Feng et al. Meshnet: Mesh neural network for 3d shape representation, https://arxiv.org/abs/1811.11424

---

### Author Response · Authors · 2024-11-27
**Revision Notes**

We thank the reviewers for their thoughtful feedback and have revised the paper to address the concerns raised and to clarify the contributions of our work. We have highlighted changes made in the updated version in red and include a brief list of updates here as well.

- Added various related works highlighted by the reviewers to the related works.
- Added additional details differentiating our work from the related works.
- Added an explanation of the relationship between the input geometry and the output spherical functions which describes the equivariant relationship between them.
- Added additional description information to Section 4 to clarify how the encoder works.
Re-wrote decoder Section 4.2 to highlight our contributions and give credit to other works, which we were not aware of at the time.
- Added additional information to the supervised learning experiments to clarify the following points:
1. We add references highlighting that G2Sphere and some baseline methods results fall within domain-specific desired/accepted error bounds.
2. Added results on the spherical CNN baseline to evaluate the effect of our learned mapping from geometry to spherical features (i.e. our encoder), and highlighted the value of our learned features.
3. Added additional details discussing Fig. 5, emphasizing the poor performance of baseline methods as a common result of using implicit models on difficult zero-shot generalization tasks.
- Other minor updates for clarity.

---

### Meta-Review · Area_Chair_mPfY · 2024-12-19

**Metareview:**

This paper introduces G2Sphere, a method for mapping object geometries to spherical signals entirely in Fourier space using equivariant neural networks. G2Sphere predicts Fourier coefficients of signals, enabling high-resolution continuous spherical signal computation. Experiments on tasks like radar response, aerodynamics drag prediction, and navigation demonstrate its superior accuracy, efficiency, and generalization compared to baselines, with benefits from equivariance and Fourier features.

Reviews for this paper are average, ranging from 5 to 6. One of the weakness highlighted by reviewers is related to weak aspects of the experimental work, and missing state-of-the-art methods. Authors improved significantly their submission in their revision. However, I believe that the amount of changes undergone might require a new round of review, and as no reviewer expressed the will to champion the paper and push it for acceptance, I am enclined to give a reject option for this paper. I encourage the authors to improve their submission by taking into account the different reviewers remarks in the discussion.

**Additional Comments On Reviewer Discussion:**

One of the weakness highlighted by reviewers is related to weak aspects of the experimental work, and missing state-of-the-art methods. Those aspects have been discussed and acknowledged by authors.

---

### Decision · Program_Chairs · 2025-01-22

Reject